# Next-Gen Healthcare Devices: Evolution of MEMS and BioMEMS in the Era of the Internet of Bodies for Personalized Medicine

**DOI:** 10.3390/mi16101182

**Published:** 2025-10-19

**Authors:** Maria-Roxana Marinescu, Octavian Narcis Ionescu, Cristina Ionela Pachiu, Miron Adrian Dinescu, Raluca Muller, Mirela Petruța Șuchea

**Affiliations:** 1National Institute for Research and Development in Microtechnologies—IMT Bucharest, 126A Erou Iancu Nicolae Street, 077190 Voluntari, Romania; octavian.ionescu@imt.ro (O.N.I.); cristina.pachiu@imt.ro (C.I.P.); adrian.dinescu@imt.ro (M.A.D.); raluca.muller@imt.ro (R.M.); 2Automation, Computers and Electronics Department, Petroleum-Gas University of Ploiesti, 100680 Ploiesti, Romania; 3Center of Materials Technology and Photonics, School of Engineering and Center of Research and Innovation (PEK), Hellenic Mediterranean University (HMU), 71410 Heraklion, Greece

**Keywords:** MEMS, BioMEMS, Internet of Bodies (IoB), personalized medicine, smart wearables, implantable sensors, biomedical microdevices, health data security, non-invasive diagnostics, Medical Internet of Things (MIoT)

## Abstract

The rapid evolution of healthcare technology is being driven by advancements in Micro-Electro-Mechanical Systems (MEMS), BioMEMS (Biological MEMS), and the expanding concept of the Internet of Bodies (IoB). This review explores the convergence of these three domains and their transformative impact on personalized medicine (PM), with a focus on smart, connected biomedical devices. Starting from the historical development of MEMS for medical sensing and diagnostics, the review traces the emergence of BioMEMS as biocompatible, minimally invasive solutions for continuous monitoring and real-time intervention. The integration of such devices within the IoB ecosystem enables data-driven, remote, and predictive healthcare, offering tailored diagnostics and treatment for chronic and acute conditions alike. The paper classifies IoB-associated technologies into non-invasive, invasive, and incorporated devices, reviewing wearable systems such as smart bracelets, e-tattoos, and smart footwear, as well as internal devices including implantable and ingestible. Alongside these opportunities, significant challenges persist, particularly in device biocompatibility, data interoperability, cybersecurity, and ethical regulation. By synthesizing recent advances and critical perspectives, this review aims to provide a comprehensive understanding of the current landscape, clinical potential, and future directions of MEMS, BioMEMS, and IoB-enabled personalized healthcare.

## 1. Introduction

The transformation of modern healthcare is being propelled by the convergence of micro- and nanoscale technologies with real-time connectivity and artificial intelligence. At the heart of this evolution are Micro-Electro-Mechanical Systems (MEMS), which enable the miniaturization of sensors, actuators, and electronic components onto compact, integrated platforms. MEMS technologies have already revolutionized several sectors, including automotive, aerospace, and consumer electronics, but their greatest impact may be in medicine, where their ability to interact precisely with biological systems enables unprecedented levels of diagnostic precision and therapeutic control.

Building upon MEMS foundations, BioMEMS (Biological MEMS) incorporate biocompatible materials, biochemical sensing mechanisms, and fluidic interfaces for direct interaction with the human body. These devices enable minimally invasive procedures, continuous physiological monitoring, and personalized treatment strategies aligned with the emerging paradigm of personalized medicine (PM). Unlike traditional healthcare models that adopt a “one-size-fits-all” approach, PM seeks to tailor interventions based on each individual’s genetic, biochemical, and physiological profile [1,2,3,4].

A key enabler of this shift is the **Internet of Bodies (IoB)**, a specialized branch of the broader Internet of Things (IoT), where human bodies become data-generating platforms through the integration of wearable, implantable, and ingestible smart devices. These IoB systems can collect, analyze, and transmit physiological data in real time, enhancing preventive care, chronic disease management, and telemedicine applications [5,6,7,8,9]. The emergence of IoB also redefines the doctor–patient relationship, turning it into a continuous digital feedback loop rather than a series of discrete interactions. If previously, this relationship was direct and exclusive, now, the Internet has become an additional participant in medical care delivery [10].

This review aims to explore the **co-evolution of MEMS, BioMEMS, and IoB** within the context of next-generation healthcare devices. We propose a functional classification of IoB technologies based on their level of invasiveness—non-invasive (external wearables), invasive (implantable or ingestible), and incorporated (embedded systems)—and analyze their integration into real-world healthcare applications such as diabetes management, cardiovascular monitoring, neurological disorders, and mobility rehabilitation.

We also discuss the underlying technologies, data architectures, and system challenges associated with these devices, including biocompatibility, power autonomy, data privacy, and regulatory compliance. Finally, the review offers critical insights into the future direction of IoB-connected medical devices, including artificial intelligence (AI)-assisted diagnostics, multi-modal sensing platforms, and personalized therapeutic feedback systems.

**Methodology of Literature Selection:** This review was designed as a narrative, critical synthesis rather than a systematic review. However, to enhance reproducibility and transparency, we provide here a brief overview of our literature selection approach. Relevant publications were identified through comprehensive searches in **Scopus**, **Web of Science**, **IEEE Xplore**, and **PubMed** databases using combinations of keywords such as “MEMS,” “BioMEMS,” “Internet of Bodies,” “personalized medicine,” “wearable sensors,” “implantable devices,” and “medical IoT.” The time window considered was primarily **2015–2025**, with earlier seminal works included where historically significant. Recent peer-reviewed syntheses provide expanded context on biomedical MEMS trends [11], wearable biosensors [12,13], and interoperability frameworks such as HL7 FHIR [14], while medical-device cybersecurity has been examined through the lens of benefit–risk assessments and IMD-specific controls [15]. Only peer-reviewed journal articles, high-impact conference proceedings, and authoritative reviews were retained, while non-academic or opinion sources were minimized and used exclusively for contextual background. Studies were included if they (i) presented original experimental or clinical results, (ii) described device architectures, fabrication strategies, or sensing principles relevant to healthcare, or (iii) discussed ethical, security, or regulatory challenges specific to IoB. This approach ensured that the review captures both foundational research and the most recent technological advances, while maintaining breadth across biomedical engineering, materials science, and clinical application domains.

By providing a comprehensive synthesis of current developments and future trends, this paper aims to support researchers, engineers, and clinicians in understanding the pivotal role of MEMS and BioMEMS in building the data-driven, patient-centric healthcare systems of tomorrow.

## 2. MEMS and BioMEMS: Fundamentals and Evolution

Micro-Electro-Mechanical Systems (MEMS) are miniature devices that integrate mechanical elements, sensors, actuators, and electronics on a common silicon substrate using microfabrication technology.

The functionality of MEMS (and BioMEMS) sensors is governed by the physical transduction mechanisms that convert external stimuli into measurable electrical signals. The most common principles include **piezoresistive**, **capacitive**, and **optical** sensing, each offering distinct advantages and limitations depending on the application.

**Piezoresistive sensing** relies on the change in electrical resistance of a material under mechanical strain. When a force (F) is applied, it induces a stress (σ) and corresponding strain (ε) in the sensing element, altering its resistivity according to:

ΔR/R = πσ, where (π) is the piezoresistive coefficient. This principle is widely used in pressure and acceleration sensors due to its simplicity and compatibility with standard CMOS processes. However, piezoresistive sensors often exhibit higher power consumption compared to capacitive alternatives.

**Capacitive sensing** detects variations in capacitance resulting from changes in distance or dielectric properties between electrodes. The capacitance (C) is expressed as: C = εA/d, where (ε) is the permittivity of the dielectric medium, (A) is the electrode area, and (d) is the separation distance. Capacitive MEMS sensors are preferred for inertial measurement units (IMUs) and pressure sensors due to their high sensitivity, low noise, and low power consumption. Their main drawback is susceptibility to parasitic capacitance and environmental noise.

**Optical sensing** leverages changes in optical path length, interference, or reflectivity to detect mechanical displacement or biochemical interactions. For example, in an optomechanical accelerometer, the displacement (x) of a proof mass modifies the resonant wavelength λ of a photonic crystal cavity: Δλ~x.

This approach offers ultrahigh sensitivity and immunity to electromagnetic interference, making it ideal for applications requiring precision, though it typically requires more complex readout systems and higher power budgets.

Figure 1 illustrates the three transduction mechanisms side by side: (**a**) piezoresistive beam bending and resistance change; (**b**) capacitive parallel-plate displacement sensor; (**c**) optical cavity displacement detection.

Signal conditioning and processing play crucial roles in MEMS and BioMEMS sensor performance. Analog front-ends often include Wheatstone bridges (for piezoresistive elements), charge amplifiers (for capacitive sensors), or photodiode transimpedance amplifiers (for optical detection). Increasingly, edge computing architectures integrate microcontrollers for on-device preprocessing, feature extraction, and power optimization before data transmission, thereby reducing latency and improving energy efficiency. Techniques such as duty cycling, event-driven sensing, and energy harvesting are commonly employed to extend operational lifetime in IoB applications.

Their **inception** in the 1980s marked a pivotal point for miniaturization in sensing and control technologies. Early MEMS were primarily based on silicon micromachining, enabling the development of accelerometers and pressure sensors with applications in automotive and industrial sectors. However, their true potential has emerged in the biomedical field, where precision, sensitivity, and low power consumption are paramount.

The evolution of MEMS can be delineated into four major developmental stages. During the **early development phase** in the 1980s and 1990s, research was centered on silicon planar technology with supplementary steps as anisotropic etching of silicon, which allowed obtaining of 3D movable structures. MEMS devices integrated mechanical 3D parts with electronics on the same chip.

Primarily were also utilizing piezoresistive and capacitive sensing elements. This era also witnessed the first implementation of rudimentary packaging techniques suitable for microscale devices. Progressing into the **2000s**, the field entered a miniaturization and integration phase, marked by significant improvements in lithography and thin-film deposition technologies. These advancements facilitated the integration of multiple sensor modalities, on-chip signal processing, and the development of system-on-chip (SoC) architectures tailored to specific applications.

The **2010s** steered in a phase of material diversification, as new classes of materials—including polymers, metals, and hybrid composites—were adopted to expand the functional and mechanical capabilities of MEMS. These materials enabled the fabrication of flexible, wearable, and even implantable devices, combining enhanced mechanical resilience with improved biocompatibility [16]. In the **current decade**, MEMS have entered a modern phase characterized by the convergence with wireless communication technologies such as Bluetooth Low Energy (BLE) and Wi-Fi, as well as integration into broader paradigms like the Internet of Things (IoT) and the Internet of Bodies (IoB). Moreover, the fusion of MEMS platforms with artificial intelligence and cloud-based analytics has opened up unprecedented possibilities for data-driven, networked sensing systems [5,6].

The field of MEMS fabrication has recently undergone significant innovation, driven by the need to overcome the rigidity, planar constraints, and slow prototyping cycles of traditional silicon-based microfabrication. Three major research directions—hybrid 3D printing, stretchable electronics, and biodegradable materials—are now reshaping the design space of microelectromechanical systems for next-generation healthcare applications. One transformative approach is hybrid additive manufacturing, which integrates microscale 3D printing techniques with conventional material deposition. For example, researchers have demonstrated fully functional MEMS accelerometers fabricated via two-photon polymerization combined with directional metal evaporation to form integrated strain gauges. These devices showed stable resonant behavior and mechanical sensitivity, while enabling highly complex three-dimensional geometries that would be unachievable using planar photolithography techniques [17]. Similarly, multi-material 3D printing strategies have been used to fabricate flexible pressure sensors that integrate polymeric insulation matrices, carbon-nanotube-based electrodes, and ionic gel electrolytes, all deposited onto a single stretchable platform. This level of integration represents a major step toward fully additive MEMS manufacturing, where both structural and functional components are printed in a single workflow [18].

In parallel, the development of intrinsically stretchable and mechanically adaptive MEMS platforms has gained momentum. Traditional rigid substrates have been replaced with soft elastomers and conductive polymer composites that accommodate large strains without loss of electrical integrity. For instance, hydrogen and pressure sensors have been successfully developed using elastic polyimide matrices embedded with conductive nanomaterials, enabling continuous monitoring under dynamic deformation [19]. Beyond material innovations, mechanical structuring strategies—such as serpentine interconnects [20,21], buckled thin films [22], and kirigami-inspired geometries [23]—have made it possible to incorporate high-performance inorganic MEMS components onto stretchable substrates. These strategies allow for strain tolerance exceeding 20%, with preserved conductivity and device function during repetitive deformation cycles, as demonstrated in pioneering work by Hilbich et al. in [24]. Another rapidly emerging trend is the development of biodegradable and transient MEMS systems. These platforms are designed to degrade safely within the body or environment after performing their function, eliminating the need for surgical retrieval or long-term biointegration. Biodegradable microdevices have been fabricated using natural and synthetic polymers such as gelatin-methacryloyl (GelMA), polylactic acid (PLA), and poly(ethylene glycol) (PEG)-modified networks. These materials offer tunable degradation profiles and mechanical properties suited for transient sensors, actuators, and microfluidic platforms [25]. In addition, bioinspired composite systems such as Aguahoja—constructed from chitosan, cellulose, and silk—exemplify the potential of combining ecological sustainability with device-level functionality, potentially enabling MEMS that are both high-performance and environmentally regenerative.

Altogether, these advancements highlight a profound shift in MEMS fabrication from rigid, lithography-based approaches toward flexible, 3D-structured, and even biodegradable systems. As these novel fabrication strategies mature and integrate with wireless communication and AI-enabled control, they are expected to unlock new classes of wearable, implantable, and transient medical devices that blur the boundary between body and machine.

The fabrication of MEMS devices relies on a broad set of technological processes, many of which originate in the microelectronics industry but are significantly extended and adapted to meet the requirements of micro-electromechanical structures. In addition to conventional photolithography, oxidation, and doping steps, MEMS manufacturing incorporates specialized techniques such as thin-film material deposition, surface and bulk micromachining, and selective etching processes tailored to the target device geometry and functionality.

Because MEMS integrate mechanical, electrical, and electronic components on a single chip, their development is inherently multidisciplinary, combining expertise in materials science, mechanical engineering, microfabrication, and electronic design. Both device design and testing demand sophisticated tools, including three-dimensional modeling, thermo-mechanical and electromagnetic simulations, and multiphysics optimization. Particular attention must be paid to potential failure modes such as stiction, thermal stress, yield strain, and long-term reliability, as well as to microscale material defects that can critically affect performance.

Manufacturing costs for MEMS devices are typically higher than those for standard semiconductor components, reflecting the additional fabrication steps, the use of materials with tailored mechanical or biocompatible properties, and the more complex encapsulation processes required. Packaging is especially challenging, as MEMS devices often cannot be housed using standard semiconductor approaches due to their moving elements and environmental sensitivity.

Nevertheless, the multidisciplinary nature of MEMS fabrication and their seamless integration with **integrated circuit (IC)** technologies enable the creation of highly advanced and intelligent microsystems. These platforms combine sensing, actuation, and signal-processing capabilities in compact architectures, offering innovative functionalities and paving the way for next-generation smart systems with broad application potential.

This technological trajectory has culminated in the emergence of BioMEMS—a specialized subdomain of MEMS designed for biomedical and clinical applications. Unlike conventional MEMS, BioMEMS devices typically interact directly with biological fluids or tissues and are constructed from biocompatible materials such as polydimethylsiloxane (PDMS), SU-8 photoresist, hydrogels, or surface-modified silicon substrates. Their design necessitates careful consideration of surface chemistry, sterilization protocols, and compatibility with living systems [5].

The fabrication of **BioMEMS** devices builds upon the core principles of MEMS manufacturing but introduces additional requirements dictated by biological interfacing, fluidic handling, and biocompatibility constraints. While photolithography, thin-film deposition, and etching remain foundational processes, BioMEMS fabrication often incorporates **soft lithography, polymer micromolding, and microfluidic channel formation** to enable precise manipulation of biological fluids and cells at the microscale. Materials such as **polydimethylsiloxane (PDMS)**, **SU-8**, **parylene-C**, and biocompatible metals or polymers are commonly employed to ensure chemical inertness, optical transparency, and mechanical flexibility suitable for biological environments.

The inherently multidisciplinary nature of BioMEMS development extends beyond traditional microfabrication, integrating **biomaterials science, surface chemistry, and biomedical engineering**. Device design frequently involves **multiphysics modeling** to account for coupled phenomena such as fluid flow, electrokinetic transport, and biomolecular interactions, while testing protocols must address sterility, cytocompatibility, and performance stability in physiological conditions.

Similar to MEMS, BioMEMS devices face critical challenges such as **biofouling, delamination, and surface degradation**, which can compromise sensitivity and long-term reliability. Encapsulation and packaging are even more complex due to the need for fluidic access, gas permeability, and sterility maintenance. These factors contribute to **higher fabrication costs** compared to conventional MEMS or semiconductor devices.

Despite these challenges, BioMEMS technologies unlock unique capabilities for **in vitro diagnostics, lab-on-chip systems, implantable biosensors, and organ-on-chip platforms**. Their seamless integration with **IC-based signal processing, wireless communication modules, and AI-driven analytics** enables the development of compact, multifunctional systems that bridge the gap between microengineering and biomedicine. As a result, BioMEMS stand at the forefront of next-generation healthcare devices, enabling real-time biological monitoring and precision medicine applications.

Although MEMS and BioMEMS share a common technological foundation rooted in microelectronics manufacturing, their fabrication approaches diverge significantly due to the distinct operational environments and functional requirements of each. **MEMS fabrication** emphasizes mechanical precision, structural stability, and integration with electronic circuits, relying primarily on silicon-based substrates, thin-film deposition, and surface or bulk micromachining to produce sensors and actuators with high spatial resolution and mechanical robustness.

In contrast, **BioMEMS fabrication** must accommodate biological interfaces, fluidic manipulation, and biocompatibility constraints. This necessitates the incorporation of **soft materials**, flexible substrates, and specialized processes such as **soft lithography, microfluidic channel etching, and surface functionalization**. Additionally, while MEMS packaging focuses on mechanical protection and electrical connectivity, BioMEMS encapsulation must ensure sterility, allow controlled fluid exchange, and maintain biointerface functionality.

These differences reflect the complementary roles of MEMS and BioMEMS technologies: MEMS devices excel in precise physical sensing and robust structural performance, whereas BioMEMS platforms extend microfabrication into the biological domain, enabling direct interaction with complex biochemical systems. Together, they form the technological backbone of modern body-integrated microsystems and play a central role in the evolving Internet of Bodies landscape.

The diversity of MEMS and BioMEMS platforms has rapidly expanded to address the increasing demand for miniaturized, energy-efficient, and high-performance sensors in personalized medicine and body-integrated electronics. A systematic comparison of representative MEMS and BioMEMS devices reveals important distinctions in sensing modality, resolution, material composition, transduction mechanisms, and technology readiness levels (TRLs). Table 1 provides a structured benchmark of key platforms across these dimensions, enabling a contextual evaluation of their clinical and technological maturity.

Classical MEMS devices are predominantly based on silicon micromachining and are well-suited for inertial, magnetic, and environmental sensing. For example, resonant piezoelectric MEMS magnetometers have demonstrated sub-nanotesla sensitivity using silicon and magnetostrictive composite materials, with operating power in the microwatt to milliwatt range, making them suitable for low-power wearable electronics 1. Similarly, cavity-enhanced optomechanical accelerometers leverage photonic crystal structures for highly sensitive mechanical displacement detection, reaching noise floors near 10 µg/√Hz in broadband applications 2. Despite their precision, these devices are still in research in order to be integrated with complex ICs. Recent advancements in nanomaterials have enabled graphene-based NEMS (nano-electro-mechanical systems), which exhibit ultra-high sensitivity and reduced mass, offering excellent potential for next-generation inertial and pressure sensors. A recent study demonstrated the feasibility of suspended graphene accelerometers with robust electrical readout and high reproducibility, albeit at TRL 4–5, indicating ongoing experimental validation 3. In contrast, BioMEMS platforms are specifically engineered to function within or in close contact with biological environments. For instance, silicon-based neural probes—such as the Michigan or Utah arrays—remain gold standards for electrophysiological signal acquisition due to their ability to resolve single-unit activity in the µV range, operating within TRL 7–9 for clinical and research use 4. These implantable systems prioritize material biocompatibility (e.g., Pt, Ir) and encapsulation stability to ensure chronic performance and minimal tissue response. Wearable biochemical BioMEMS devices, such as microfluidic sweat sensors, have gained prominence for non-invasive monitoring of electrolytes and metabolites. These devices, typically fabricated using PDMS or PET substrates integrated with carbon nanotube or metal oxide electrodes, employ amperometric or potentiometric readout for real-time physiological feedback. A representative system demonstrated reliable sodium ion detection in sweat with low power requirements (<1 mW) and TRLs in the range of 5–7, highlighting their near-market readiness for sports and health applications 5.


**Quantitative benchmarking and implications for device selection**


The inclusion of quantitative parameters such as sensitivity, power consumption, and technology readiness levels (TRL) offers a clearer perspective on the trade-offs inherent in MEMS and BioMEMS design. For instance, resonant piezoelectric MEMS magnetometers achieve sensitivities below 1 nT while consuming only microwatt-level power, making them ideal for wearable applications where continuous operation is critical. Conversely, optomechanical accelerometers achieve superior resolution but at the cost of increased optical power, which limits their suitability for battery-constrained devices. Graphene NEMS accelerometers push the sensitivity frontier further but remain at experimental TRLs, highlighting ongoing material and fabrication challenges.

On the biological side, neural probes demonstrate mature clinical performance but require surgical implantation, while wearable sweat sensors achieve lower sensitivity but offer non-invasive operation and user comfort. These examples illustrate that **device performance cannot be evaluated solely by sensitivity or resolution**; rather, it must be contextualized in terms of power efficiency, invasiveness, and readiness for clinical deployment.

Collectively, this comparative analysis underscores several key trends: (i) MEMS devices excel in mechanical and inertial sensing with high miniaturization and signal fidelity; (ii) BioMEMS platforms extend functionality into biochemical and electrophysiological domains with superior biocompatibility; and (iii) power autonomy and material integration remain critical bottlenecks across both domains. As MEMS and BioMEMS continue to converge with wireless communication, artificial intelligence, and bioresorbable electronics, their impact on the future of personalized medicine is poised to expand significantly [26,27,28,29,30].

BioMEMS platforms support a broad array of diagnostic and therapeutic applications. They are central to point-of-care testing (POCT) technologies, enabling the detection of biomarkers from minimal sample volumes in decentralized or resource-limited environments [31,32]. In parallel, implantable sensors and stimulators based on MEMS technology—such as those used for continuous glucose monitoring (CGM), cardiac telemetry, or targeted neural stimulation—are transforming chronic disease management and personalized therapy [33]. Additionally, microfluidic and lab-on-chip (LOC) systems have emerged as integral components of BioMEMS, allowing complex biochemical operations, including sample preparation, reagent mixing, and biosensing, to be miniaturized and integrated on a single substrate [11].

The rise of BioMEMS strongly aligns with the overarching objectives of personalized medicine, which aims to provide diagnostic and therapeutic strategies tailored to an individual’s genetic profile, behavior, and environment [2,3,34]. By enabling minimally invasive, continuous monitoring and offering real-time physiological feedback, BioMEMS technologies serve as the technological foundation for proactive health management and precision clinical decision-making.

Furthermore, the integration of BioMEMS with **IoB architectures** creates powerful tools for managing chronic diseases such as diabetes, cardiovascular conditions, and respiratory disorders [35]. For example, glucose sensors connected to insulin pumps enable real-time insulin titration, while wearable ECG patches linked to cloud platforms allow remote arrhythmia monitoring [36,37]

In summary, MEMS and BioMEMS technologies have evolved from passive microstructures into active, intelligent biomedical systems. Their fusion with connectivity and computation through the IoB paradigm marks the beginning of a new era of personalized, anticipatory, and highly adaptive medicine.

A comparative analysis of MEMS and BioMEMS reveals complementary strengths and persistent trade-offs. **MEMS platforms** excel in mechanical, inertial, and magnetic sensing due to their high precision, miniaturization potential, and low power consumption. However, their interaction with biological media is limited, and they often require encapsulation or surface modification for biocompatibility. **BioMEMS devices**, on the other hand, are optimized for biochemical and electrophysiological sensing and are inherently designed to operate within biological environments. Their advantages include direct tissue interface, real-time biochemical monitoring, and compatibility with fluidic systems. The limitations differ accordingly: MEMS devices face challenges in biological integration and packaging, whereas BioMEMS systems must address biofouling, long-term stability, and sensitivity drift. Furthermore, BioMEMS platforms often consume more power due to electrochemical transduction and signal conditioning. Future device generations must bridge these differences by combining the structural precision of MEMS with the biocompatibility and functional specificity of BioMEMS. Despite the impressive progress achieved over the past two decades, MEMS and BioMEMS technologies continue to face fundamental limitations that hinder their full clinical translation. A primary challenge lies in the **trade-off between miniaturization and functional performance**. While reducing size improves biocompatibility, patient comfort, and minimally invasive deployment, excessive miniaturization can compromise sensing resolution, signal-to-noise ratio, power storage, and communication bandwidth. Achieving an optimal balance between device footprint and functionality remains an active area of research. Another key limitation concerns **long-term biostability and biocompatibility**. Even materials considered biocompatible may undergo corrosion, delamination, or biofouling when exposed to physiological environments for extended periods. This leads to sensor drift, degraded signal quality, and reduced operational lifetimes. Encapsulation strategies and surface functionalization approaches are being investigated to mitigate these issues, but a universally reliable solution is still lacking. Finally, **interoperability and standardization** represent systemic barriers to clinical adoption. MEMS and BioMEMS devices are often developed in isolated research contexts without standardized communication protocols, data formats, or regulatory pathways. As a result, integrating these devices into larger medical Internet-of-Things (MIoT) infrastructures remains challenging. Bridging this gap will require concerted efforts to define interoperable interfaces and regulatory frameworks tailored to body-connected systems.

## 3. Internet of Bodies and Medical Device Integration

The **Internet of Bodies (IoB)** represents a rapidly expanding subset of the Internet of Things (IoT) in which networked devices are integrated directly with or in close proximity to the human body. These smart biomedical systems collect, analyze, and transmit physiological or behavioral data, enabling real-time interaction between the human organism and digital infrastructures [5,6,7]. This concept expands beyond traditional wearable health technologies by incorporating implantable, ingestible, and embedded devices that interact intimately with biological systems.

In the IoB ecosystem, **MEMS and BioMEMS technologies serve as the physical interface** between the human body and the digital world. Devices can detect diverse signals—ranging from heart rate, glucose levels, respiratory function, and neural activity—to biochemical markers in bodily fluids. These signals are converted into digital data, locally processed, and transmitted wirelessly to cloud-based platforms where they are analyzed, visualized, or acted upon by healthcare providers or automated algorithms [5,32,38,39].

### 3.1. IoB System Architecture

A typical IoB system consists of the following interconnected layers:**Sensing layer**: embedded MEMS/BioMEMS sensors capture raw physiological data (e.g., temperature, pressure, motion, biochemical composition).**Communication layer**: short-range wireless protocols (e.g., Bluetooth Low Energy (BLE), Wi-Fi, and **narrowband Internet of Things (NB-IoT)** enable data transmission to smartphones, gateways, or cloud platforms [5,40].**Processing and analytics layer**: on-board microcontrollers perform initial data filtering. Cloud-based analytics—often supported by AI—identify patterns, trigger alerts, or optimize treatment plans [41].**Application layer**: interfaces for physicians, patients, and caregivers provide dashboards, alerts, and decision support tools to act upon the processed health information.These systems support both **real-time diagnostics** and **longitudinal monitoring**, offering clinicians a continuous and contextualized view of a patient’s health status—unachievable with conventional episodic care

### 3.2. Cross-Domain Context of IoB: From IoT to Human Integration

The IoB is part of a broader family of IoT systems applied across multiple domains. Table 2 outlines how different sectors—such as agriculture, oceanography, space, and defense—use sensor-based connectivity, contextualizing IoB as the human-facing frontier of this technological. Although examples such as non-ground aerospace networks and underground soil sensing may appear peripheral, they are included here to illustrate how IoB architectures are increasingly influenced by IoT paradigms developed in adjacent domains. These cross-domain insights often translate into healthcare innovation, particularly in areas like autonomous system coordination, ultra-low-power communication, and distributed sensor networks.

### 3.3. IoB Device Classification and Interaction with the Human Body

Internet of Bodies (IoB) devices can be categorized according to their level of physical interaction with the human body, a framework that is essential for guiding risk assessment, regulatory classification, and interface design strategies [8,56]. At the least invasive level, **non-invasive external devices** include wearables such as smartwatches, rings, bracelets, textile-integrated sensors, smart shoes, and virtual reality headsets. These platforms operate on the body’s surface and are currently the most widely adopted due to ease of use and minimal safety concerns [5,52]. Moving deeper into the body, **invasive internal devices** comprise implants (such as pacemakers and neural stimulators), ingestible technologies (like digital pills and gastrointestinal cameras), and injectable biosensors designed for continuous physiological monitoring [33,35,40]. The most advanced category encompasses **incorporated devices**, which integrate bioelectronic systems directly into the body’s architecture. Examples include brain–computer interfaces, bionic limbs, and smart prostheses that not only monitor but also actively augment biological functions [56,57]. While these classifications are conceptually distinct, many emerging IoB solutions blur these boundaries through hybrid interaction modes and multi-modal sensing configurations tailored for personalized medicine and continuous digital healthcare. We outline here the principles that govern IoB layering—how sensing depth and bidirectionality shape requirements for power, packaging, and data protection. Concrete device categories and examples are consolidated in Section 5.1, which maps representative platforms to this framework to avoid repetition.

### 3.4. IoB Connectivity and Data Flow


**Internet of Bodies: A Multi-Layered Ecosystem of Health Monitoring Devices**


As it is presented by Celik A. et al. in Ref. [5], different data are collected by IoB, from four big categories: Healthcare, Wellness/Fitness, Sports and Entertainment (Figure 2a) and a schematic representation of the multi-layer data flow in the Internet of Bodies (IoB) ecosystem. MEMS and BioMEMS sensors acquire physiological, biochemical, and environmental data, which undergo on-device preprocessing before transmission via wireless communication networks. Cloud-based platforms perform large-scale analytics and integrate the results into clinical decision support systems. The feedback loop enables closed-loop therapeutic control, ensuring adaptive and personalized medical interventions as shown in Figure 2b.

IoB represents a multi-tiered framework in which medical devices—classified by their degree of bodily integration—are connected through digital networks for health data acquisition, monitoring, and control. IoB applications that combine “Healthcare” and “Sports” were largely discussed in Ref. [58]. Here are presented wearable sensor devices for prevention and rehabilitation in healthcare: swimming exercise with real-time therapist feedback. The device monitors the high-intensity swimming for different number of laps with different swimming styles. The recorded data from both swimming tests offers information about the variation in swimming intensity, the swimming rhythm. It is used to evaluate the swimming motion dynamics of the lower back. Focusing attention on the first category from the “Healthcare” sector, PM based on IoB is a big advantage for the world because it has the potential to help patients more efficiently and effectively than traditional medicine [59]. Healthcare systems generate a huge volume of different types of data. Due to the complexity of the medical information, it is hard to create a general pattern considering all the aspects of health. So, regarding the fact that each patient has its special needs, and different problems, the devices are focused on collecting the essential data, the needed data. To collect data from the human body, the IoB devices (healthcare monitoring devices) are used. The evolution of Wearable Devices with Real-Time Disease Monitoring for PM happened gradually from accessories, integrated clothing, body attachments and finally through body inserts [33]. There are **three categories of devices connected to *IoB*** that are found **in**, **on**, or **around** the human body. The main two categories are **invasive** and **non-invasive** devices, based on their location.

This layered categorization is visually synthesized in Figure 3, which outlines the three principal groups of IoB devices: **non-invasive (external)**, **invasive (internal)**, and **incorporated (integrated)**. Each group includes numerous device types with progressively increasing levels of physical integration and interaction with the human body. Non-invasive devices such as smartwatches, biosensor-equipped tattoos, and smart shoes dominate the current commercial landscape, while internal solutions—such as ingestible sensors and pacemakers—are becoming more sophisticated and patient-friendly. Fully incorporated technologies, like bionic eyes and brain–computer interfaces (BCIs), exemplify the frontier of bio-digital convergence. As illustrated, the interconnection of these devices across layers supports a complex data flow that feeds into PM, early diagnostics, remote monitoring, and AI-driven clinical decision-making. This multi-scale, body-centric connectivity forms the structural backbone of the IoB ecosystem as discussed in the following sections.

At the system level, data captured from MEMS and BioMEMS sensors are typically preprocessed locally (e.g., through edge computing or on-device microcontrollers), and then transmitted securely via short-range wireless protocols (Bluetooth Low Energy, Wi-Fi, NB-IoT) to smartphones, gateway hubs, or cloud platforms. In the cloud, real-time analytics and AI-based algorithms interpret this data to detect anomalies, personalize interventions, or support clinician decision-making [5,40,57].

This connectivity loop closes through feedback: alerts, therapy adjustments, or emergency interventions are delivered back to the patient or caregiver in real time. The overall process—from sensor acquisition to clinical action—is illustrated schematically in Figure 4, which captures the typical IoB data acquisition pipeline and stakeholder flow. Data captured from smart biosensors is typically preprocessed locally, transmitted securely via encrypted channels, and then stored and analyzed in a protected cloud or edge-computing environment. From here, it may be shared with authorized healthcare personnel or used to trigger emergency responses, adjust therapies, or visualize trends on patient dashboards.

While IoB platforms promise continuous, data-driven healthcare, their implementation faces several unresolved challenges. **Interoperability gaps** between devices from different manufacturers complicate seamless data aggregation and clinical integration. **Data overload and algorithmic bias** pose additional risks, as AI-based analytics trained on limited or unrepresentative datasets may generate misleading inferences. Moreover, **cybersecurity vulnerabilities**—including potential wireless interception or device hijacking—represent critical safety concerns, especially for life-sustaining implants. It is also worth noting that reported device performance metrics often vary widely between laboratory prototypes and real-world clinical use. For example, many biosensors demonstrate excellent sensitivity in vitro but suffer from drift, encapsulation, or immune response effects in vivo. Such discrepancies underscore the need for **standardized testing protocols** and large-scale clinical validation to bridge the gap between experimental success and reliable patient outcomes. **Comparative insights: IoB integration challenges:** When MEMS and BioMEMS technologies are integrated into IoB architectures, performance trade-offs become more pronounced. **Non-invasive devices** are advantageous for patient compliance and rapid deployment but often suffer from signal attenuation and variability due to skin impedance or environmental interference. **Implantable and ingestible devices** deliver high-fidelity data and continuous monitoring but face significant hurdles in terms of surgical deployment, long-term safety, and regulatory approval. **Incorporated devices**, including brain–machine interfaces, offer transformative capabilities but remain at low TRL levels and pose unresolved ethical and safety challenges. At the system level, **power autonomy, secure data transmission, and interoperability** are recurring bottlenecks across all categories. Solutions such as energy-harvesting power systems, standardized communication protocols, and advanced encryption strategies are actively being explored but require further validation before widespread clinical use.

## 4. Cases of MEMs and IoB in Personalized Medicine

The integration of MEMS, BioMEMS, and IoB technologies into clinical applications has transformed healthcare from reactive to proactive. Continuous real-time monitoring, early diagnostics, and adaptive therapy are now achievable through wearable, implantable, and ingestible devices tailored to individual physiology. This section explores key use cases in PM enabled by IoB systems, categorized by medical domain and patient need. The evolution of Wearable Devices with Real-Time Disease Monitoring for PM happened gradually from accessories, integrated clothing, body attachments and finally through body inserts [33].

### 4.1. Diabetic Monitoring and Smart Orthotic Systems

Diabetes mellitus is a chronic metabolic disorder that affects hundreds of millions of individuals globally, imposing a substantial healthcare burden due to its long-term complications and need for continuous management. Effective self-care includes tight glycemic control and proactive foot monitoring to prevent sequelae such as retinopathy, neuropathy, cardiovascular damage, and diabetic foot ulcers. To address these challenges, MEMS-based technologies have played a transformative role in revolutionizing diabetes care. Devices such as **continuous glucose monitors (CGMs)**, **wearable insulin pumps**, and **smart orthotic systems** represent some of the most successful early integrations of biomedical microdevices into daily clinical practice. These tools leverage microfabricated sensors, wireless communication, and data analytics to provide real-time, patient-specific feedback and therapeutic guidance.

Among these, **glucose sensing** remains one of the most mature and widespread applications within the IoB domain. Traditional capillary blood glucose testing—typically involving finger-prick methods—has increasingly been replaced by continuous and non-invasive alternatives. The **Eversense^®^ CGM**, for instance, employs an **implantable fluorescence-based glucose sensor** placed subcutaneously in the upper arm [60]. It can remain functional for up to six months and communicates wirelessly via a **wearable transmitter** adhered over the sensor site. The transmitter relays data to a **smartphone application**, allowing users to monitor glucose trends in real time, with alerts for hypoglycemic and hyperglycemic events. In parallel, **wearable optical sensors** utilizing **reflectance-based photoplethysmography (PPG)** in the **red and near-infrared (R-NIR) spectral regions** are being developed to estimate blood glucose concentration through the skin. These devices take advantage of glucose’s influence on optical absorption and scattering properties of tissues, providing a non-invasive, calibration-driven estimation of glucose dynamics without penetrating the skin barrier [33].

Beyond blood and interstitial fluid, researchers have explored **alternative biofluids** for glucose monitoring, including **saliva**, which offers an accessible and painless sampling medium. Arakawa et al. designed a **mouthguard-type biosensor** that integrates **platinum and Ag/AgCl electrodes** encapsulated in a **PDMS membrane**. The device allows for the detection of glucose in saliva using amperometric techniques, demonstrating a promising pathway toward **discreet, non-invasive, and wearable biochemical monitoring** [61]. Such developments exemplify the expanding scope of BioMEMS platforms beyond traditional sampling strategies.

In addition to glucose monitoring, the prevention of **diabetic foot ulcers**—a severe and often under-addressed complication resulting from **peripheral neuropathy and impaired circulation**—has emerged as a critical use case for IoB-based foot-care technologies. Innovative solutions include **smart orthotics** and **sensor-embedded insoles** that enable real-time gait and pressure analysis. Dabiri et al. [35] proposed a **MEMS-integrated electronic orthotic shoe** capable of monitoring plantar pressure distribution and gait abnormalities. The system continuously captures biomechanical data and provides **real-time alerts** to the patient, prompting adjustments in posture or activity to reduce stress concentrations and prevent tissue breakdown. Complementarily, **3D-printed insole platforms** embedded with **flexible temperature and pressure sensors** have been designed to redistribute plantar load. These devices detect **thermal hotspots** and **abnormally high localized pressures**, both of which are key precursors to ulcer formation, and can communicate wirelessly with mobile devices to alert the user or healthcare provider [62,63].

Collectively, these advancements in IoB and BioMEMS technologies provide a **closed-loop system** for diabetes management—integrating sensing, feedback, and therapeutic action. They illustrate how smart, body-integrated microdevices can enhance self-care, reduce complication rates, and support proactive, data-driven medical decision-making in chronic disease contexts.

### 4.2. Cardiovascular and Respiratory Monitoring

BioMEMS-enabled wearables have become pivotal in the continuous monitoring of cardiovascular and respiratory parameters, offering new frontiers in the management of chronic conditions such as arrhythmias, hypertension, and heart failure. In cardiology, the integration of MEMS-based sensors into **wearable platforms**—including smart bracelets, textile-based patches, and flexible skin-conformal electronics—has enabled **real-time tracking of key physiological metrics** such as electrocardiogram (ECG) signals, heart rate variability, peripheral oxygen saturation (SpO_2_), and even cuffless blood pressure estimation. These systems utilize embedded MEMS pressure sensors, optical detectors, or capacitive electrodes to capture dynamic biosignals and wirelessly transmit alerts via connected mobile applications or cloud platforms [52,53]. A medical wristband connected to a website that enables real-time viewing of all the data provided by the bracelet’s sensors (body temperature, heart rate, medication intake, and abnormal shaking) was presented by Hummady et al. in [64]. WHOOP Strap or Oura Ring are other two examples of such smart devices that monitor health [65,66].

A particularly impactful application of BioMEMS in cardiology is the **next-generation pacemaker**. Traditional cardiac pacemakers are powered by internal batteries that require periodic surgical replacement, posing infection risks and increasing long-term healthcare costs. Recent advances have introduced **battery-free pacemakers**, which derive power from **radiofrequency (RF) electromagnetic fields** or **thermoelectric energy harvesting** from the body’s own heat gradients. These systems incorporate ultra-low-power MEMS circuits and wireless energy receivers to regulate cardiac rhythm without bulky internal batteries, thereby reducing device size and extending operational lifespan.

Umer M. et al. investigated such a device for health monitoring using IoT and AI-based solutions. Pacemakers can generate problems like the battery, electromagnetic interference, and infection. Zheng Q. et al. studied the battery-free pacemaker technology, that can eliminate the need for battery replacement surgeries of traditional pacemakers. Sun et al. made a battery-less mm-sized wirelessly powered pacemaker microchip with on-chip antenna in 180nm CMOS process. The microchip harvests RF radiation from an external source in the X-band frequency, with the size of 4mm by 1mm. The in vivo experiment is demonstrated successfully on a live pig heart [67,68,69].

In the context of respiratory health, BioMEMS technologies have enabled **non-invasive, wearable solutions** for detecting abnormalities in breathing patterns. Devices equipped with **MEMS accelerometers**, **strain gauges**, or **triboelectric nanogenerators (TENGs)** can continuously monitor **respiratory rate**, **cough frequency**, and **chest wall motion**. These parameters are essential for diagnosing or managing **chronic obstructive pulmonary disease (COPD)**, **asthma**, and **obstructive sleep apnea**, especially in aging populations and high-risk patients [70]. Moreover, during the COVID-19 pandemic, BioMEMS-equipped **smart wristbands** and patches were rapidly deployed to track **skin temperature variations**, **pulse oximetry**, and even **contact history**, functioning as digital sentinels for early outbreak detection and epidemiological surveillance [71,72,73,74,75,76]. One useful bracelet was made by Fang B. et al. doing the pandemic period of COVID-19. They developed a smart bracelet that not only measured the body temperature but also the analyzed the movement tracking, in order to avoid a possible infected person. These bracelets were recommended by the researchers for government agencies, schools, and aggregated office spaces [77].

Together, these cardiopulmonary BioMEMS platforms reflect a shift from intermittent clinic-based assessments to **ubiquitous, real-time health monitoring**, bridging the gap between diagnostics and preventive care. Their integration into the broader IoB infrastructure reinforces a future vision of **personalized, proactive medicine**, powered by low-power microsystems and intelligent analytics.

### 4.3. Neurophysiological and Sleep Monitoring

MEMS and IoB-integrated systems have significantly advanced the non-invasive monitoring of brain activity, motor control, and neuromuscular function, particularly through the development of **wearable and epidermal bioelectronic interfaces**. One emerging class of these systems involves **electronic tattoos (e-tattoos)**—ultrathin, flexible electronics fabricated from biocompatible materials such as **graphene**, **silk fibroin**, or **carbon nanotube (CNT)-based composites**. These skin-conformal devices can adhere to the forehead, scalp, or facial skin to continuously monitor **electroencephalogram (EEG)** and **electromyogram (EMG)** signals, alongside auxiliary parameters such as skin hydration, muscle activation patterns, or even subtle facial expressions [78]. Their mechanical flexibility, low modulus, and minimal invasiveness make them ideal for long-term neuromonitoring in both clinical and home settings.

In parallel, **ear-EEG devices** have emerged as a promising alternative to conventional EEG headsets. These compact systems, designed to fit discreetly behind or within the auricle, allow for **unobtrusive acquisition of cortical activity**, enabling applications such as **sleep tracking**, **seizure detection**, and **neurofeedback training**. Unlike traditional EEG arrays, which are often bulky and require conductive gels, ear-EEG systems can operate in real-world environments with minimal user discomfort [79,80,81,82,83].

The signals captured by these neuro-sensing platforms are transmitted to **cloud-based analytical frameworks** using low-power wireless protocols. There, they undergo noise reduction, feature extraction, and trend analysis using AI-assisted tools. This infrastructure supports **long-term cognitive and behavioral monitoring**, with practical implications for **epilepsy management**, **neurodegenerative disease surveillance**, and **sleep quality optimization**. Collectively, these MEMS-enabled neurodevices represent a paradigm shift toward **seamless brain–cloud interfaces**, offering a foundation for personalized neurology and brain–machine integration.

### 4.4. Smart Shoes and Mobility Enhancement

MEMS-based smart footwear has emerged as a vital tool in the domains of rehabilitation, elderly care, and the early detection of neuromuscular decline. These systems integrate **pressure sensors**, **inertial measurement units (IMUs)**, and **temperature sensors** within the shoe sole or insole structure to enable **real-time assessment of gait dynamics, balance, and posture stability**. In particular, they provide valuable insights for high-risk populations such as the elderly or individuals living with **Parkinson’s disease**, **multiple sclerosis**, or those recovering from orthopedic surgery [84,85]. By capturing plantar pressure distribution, stride variability, and motion trajectories, these embedded MEMS devices facilitate continuous biomechanical monitoring in real-world environments.

Telfer S. et al. studied in [62] some personalized foot orthoses with embedded sensors that measure skin temperatures. The activity monitoring was used to estimate energy expenditure. Others, like Singh S. et. Mali Harlal [63] have studied the 3D-printed orthotic insoles for foot rehabilitation, because it was demonstrated that the redistribution of plantar pressure is the key to preventing foot ulcers and amputations.

Dabiri F. et al. [35] have developed a wireless electronic orthotics composed of lightweight embedded systems and non-invasive sensors which can be used by diabetic patients. The system monitors feet motion and pressure distribution beneath the feet in real time and classifies the state of the patient. This way, they can detect the conditions that could potentially cause a foot ulcer, and they can prevent it by having the continuous feedback mechanism, that in case of an undesired behavior or condition a pre-emptive message wirelessly to the patient and the patient’s caregiver [52]

The management of diabetes-related complications involves a multidisciplinary team comprising a diabetologist, surgeon, orthopaedist, radiologist, and dietitian. Patients need to visit these specialists periodically based on their individual needs and undergo specific analyses.

Several commercial platforms have been developed to operationalize these functionalities in user-friendly formats. For instance, the **Sensoria Smart Insoles** system is designed for **diabetic foot health management** and **fall prediction**, leveraging textile-integrated pressure sensors and connectivity with mobile applications for feedback and clinician alerts. **FeetMe** and **Plantiga** offer smart insole solutions focused on **gait analysis** in patients with **neurodegenerative conditions** or those undergoing **post-surgical rehabilitation**, enabling clinicians to track recovery progress or detect early signs of functional decline. Another example, **E-Vone Smart Shoes**, integrates MEMS-based accelerometers and gyroscopes to deliver **real-time fall detection**; once a fall is identified, the system autonomously transmits **emergency alerts to caregivers or medical personnel** via cellular or Bluetooth connections [33].

These intelligent footwear platforms not only enhance patient autonomy and quality of life but also serve as **preventive healthcare tools**, reducing the incidence of fall-related injuries—a leading cause of hospital admissions among older adults. By enabling **longitudinal gait tracking**, **personalized rehabilitation feedback**, and **early warning systems**, smart shoes exemplify the convergence of MEMS, IoB, and digital medicine in supporting independent living and proactive clinical intervention. Smart footwear solutions vary widely in sensing modality, data fidelity, and clinical applicability as one can see in Table 3. Systems such as E-Vone prioritize fall detection and safety monitoring, whereas research-grade platforms integrate pressure mapping and gait analysis for early diagnosis of neurological disorders. The diversity of these approaches illustrates how MEMS-based sensor integration enables both consumer health applications and clinically relevant diagnostic tools.

### 4.5. Summary of Use Cases and Technology Mapping

To contextualize the diversity of applications, Table 4 presents a comparative overview of representative IoB-enabled MEMS devices categorized by medical condition, device type, sensing mechanism, invasiveness, and connectivity.

Across domains such as diabetes care, cardiology, neurology, and mobility enhancement, MEMS- and IoB-enabled devices demonstrate clear clinical potential but also domain-specific challenges. Glucose sensors and orthotic systems highlight the maturity of non-invasive and minimally invasive monitoring platforms, yet accuracy can be influenced by biological variability and environmental conditions. Cardiac pacemakers show how energy harvesting and wireless powering can address longevity issues but introduce new reliability and safety considerations. Neuro-sensing platforms (e-tattoos, ear-EEG) excel in patient comfort and continuous monitoring but must overcome low signal-to-noise ratios and data interpretation complexity. These examples underscore a fundamental insight: technological maturity varies widely across applications, and successful clinical translation depends not only on device performance but also on integration into clinical workflows, regulatory pathways, and patient lifestyles.

## 5. Device Taxonomy and Technology Readiness

To systematically assess the landscape of IoB-enabled healthcare systems, devices can be classified based on their interaction modality with the human body and technological maturity. This framework aids in understanding design requirements, user risks, regulatory implications, and scalability potential.

### 5.1. Classification by Interconnection Modality

As detailed in Section 3.3, IoB devices are broadly categorized based on their level of physical interaction with the human body into **non-invasive**, **invasive**, and **incorporated** systems. Table 5 below consolidates this classification and highlights representative examples, key characteristics, and major design challenges for each category. This synthesis complements the earlier conceptual discussion by emphasizing the **engineering and regulatory considerations** that accompany increasing levels of bodily integration.

This integrated classification emphasizes the progressive trade-offs between invasiveness, signal fidelity, and user safety across IoB technologies, forming a foundational framework for regulatory assessment and clinical translation.

### 5.2. Technology Readiness and Clinical Adoption

While MEMS technologies have achieved widespread integration in consumer electronics, their clinical adoption through Internet of Bodies (IoB) systems remains uneven, influenced by factors such as device type, intended medical application, and regulatory complexity. A useful framework for assessing this translational trajectory is the **TRL** scale, which quantifies the maturity of a technology from concept to deployment. **Non-invasive wearable devices**, including smartwatches, continuous glucose monitoring patches, and fitness trackers, are typically positioned at **TRL 8 to 9**, reflecting their high level of technological maturity, large-scale commercial deployment, and—in some cases—regulatory clearance by authorities such as the U.S. Food and Drug Administration (FDA) [93,94]. In contrast, **implantable and ingestible devices**, such as digital pills and wireless biosensors, generally fall within the **TRL 5 to 7** range. These technologies are undergoing active clinical validation but have achieved only limited regulatory approval thus far. More experimental are the **embedded and neuro-integrated systems**, including brain–computer interfaces and subdermal bioelectronics, which remain at **TRL 3 to 5**, often confined to **preclinical studies** or early-phase trials. Figure 5 presents a schematic representation of TRL for MEMS technologies into IoB devices.

To support the integration of these emerging technologies into healthcare, **regulatory bodies** including the Food and Drug Administration (FDA) (U.S.), EMA (European Union), and PMDA (Japan) are developing formalized frameworks for the evaluation of digital health products. These frameworks increasingly accommodate novel features such as **artificial intelligence integration**, **continuous physiological monitoring**, and **remote data analytics**, which are core to many IoB solutions [95,96]. Nonetheless, widespread deployment of clinically impactful IoB technologies requires careful navigation of several critical challenges. First, **robust clinical validation** through large-scale, randomized controlled trials is necessary to ensure safety and efficacy. Second, **long-term biocompatibility studies** must be conducted to assess the material interactions and degradation behavior of implanted or skin-contact devices. Finally, the systems must demonstrate **cybersecurity compliance** and adhere to rigorous **ethical standards for data privacy and informed consent**, especially as they handle sensitive biometric and behavioral data.


**Summary and core challenges by device category**


The classification of IoB devices by interconnection modality reveals a clear trade-off between invasiveness, signal quality, user acceptance, and regulatory complexity. Non-invasive systems dominate current markets due to ease of use and safety but are limited by sensitivity and signal fidelity. Implantable devices achieve superior performance but must resolve challenges related to biocompatibility, infection risk, and long-term reliability. Incorporated technologies remain largely experimental, facing ethical and power supply issues that require multidisciplinary solutions. Bridging these gaps demands coordinated progress in **materials science, miniaturization, power management, and data security** to enable seamless, safe, and effective body-integrated healthcare systems.

## 6. Security, Privacy and Regulatory Landscape

The transformative capabilities of IoB systems—ranging from real-time patient monitoring to autonomous therapeutic response—are inextricably tied to their ability to acquire, transmit, and process sensitive health data. However, this data-centric infrastructure introduces critical security and privacy vulnerabilities, particularly as devices become increasingly invasive and integrated into the human body.

### 6.1. Data Privacy and Ethical Concerns

IoB devices continuously capture vast amounts of **sensitive personal health data**, encompassing not only physiological parameters such as vital signs but also **behavioral patterns, emotional states, and geolocation data**. This multidimensional information enables predictive analytics that can infer future health risks, cognitive decline, or even psychological trends. However, the collection, transmission, and storage of such data introduce significant **ethical, legal, and societal concerns**. Among the most pressing risks are the potential for **health-based discrimination**, where insurers or employers might use biometric data to profile individuals, or deny services based on inferred vulnerabilities. Additionally, **targeted manipulation** may arise when behavioral traits—such as stress levels, sleep quality, or emotional responses—are exploited to influence decisions or market specific interventions. In more advanced systems, particularly those involving **implanted, AI-driven, or neuro-integrated technologies**, there exists the further threat of **loss of personal autonomy**, where users may be subjected to continuous surveillance or algorithmic decision-making with limited oversight [97,98,99].

To address these risks, data protection frameworks such as **the General Data Protection Regulation (GDPR)** in the European Union and the **Health Insurance Portability and Accountability Act (HIPAA)** in the United States have been enacted to govern aspects of **data ownership, transparency, user consent, and access control**. These regulations mandate that users must be informed about what data are collected, how they are used, and who has access. They also emphasize **data minimization and purpose limitation**, essential principles for safeguarding personal information in health technologies. Nevertheless, it is increasingly recognized that these legal frameworks were **not originally conceived with the complexity of IoB ecosystems in mind**. The dynamic, cloud-connected, and often cross-border nature of IoB data streams challenges the static boundaries of existing laws and may necessitate future amendments to account for **continuous monitoring**, **AI-driven analytics,** and **multi-modal sensor fusion** [100,101,102,103,104].

### 6.2. Cybersecurity Threats to Body-Connected Systems

A defining and particularly critical aspect of IoB technologies is that they extend the **digital attack surface** directly onto the human body, making cybersecurity not just a matter of data protection, but a **potentially life-or-death concern**. Unlike conventional information systems, where breaches typically result in data loss or service disruption, a cyberattack on an IoB device—such as a **pacemaker**, **insulin pump**, or **neuromodulation implant**—could induce **immediate physiological harm**. For example, unauthorized access to a connected insulin pump might lead to the delivery of a harmful overdose, while interference with a cardiac pacemaker could suppress or misfire life-sustaining stimulation pulses [105,106].

The spectrum of cybersecurity threats to IoB systems includes **device hijacking**, where malicious actors gain control over therapeutic functions; **spoofing and replay attacks** that target wireless communication protocols to mimic or intercept commands; **cloud-level vulnerabilities**, where large volumes of personal health data may be intercepted, altered, or sold; and more sophisticated risks such as **AI model poisoning**, in which adversarial manipulation of training datasets can bias clinical decision-making algorithms, leading to inappropriate diagnostics or treatments [107,108,109,110,111].

To defend against such multifaceted threats, a comprehensive **security-by-design** approach is imperative throughout the IoB development pipeline. This includes implementing **end-to-end encryption (E2EE)** for all data streams, ensuring that sensitive information remains protected during transmission and storage. At the device level, **hardware-based authentication methods**—including **biometric access control** or **embedded secure elements**—add an additional layer of protection against physical or remote tampering. Furthermore, **blockchain technologies** are being explored for establishing **tamper-resistant audit trails** of electronic health records, enabling transparent and traceable access logs in decentralized healthcare environments [112,113,114,115,116,117]. Equally important is the ability to perform **secure over-the-air (OTA) firmware updates**, which allows for rapid patching of software vulnerabilities without requiring device removal or clinical visits.

As IoB systems become increasingly autonomous and interconnected, cybersecurity must evolve beyond traditional paradigms to ensure **human safety, system integrity, and patient trust**. Ongoing collaboration between engineers, clinicians, ethicists, and regulators will be crucial in developing **resilient IoB infrastructures** capable of withstanding emergent threats in an era of hyper-connected health [118,119].

### 6.3. Regulatory Landscape and Compliance Strategies

As medical technologies become increasingly digitized, miniaturized, and connected, regulatory bodies worldwide are actively formulating **specialized frameworks** to govern the safe and effective deployment of IoB and MIoT systems. In the United States, the **FDA** has established the **Digital Health Center of Excellence**, which provides dedicated guidance for cybersecurity preparedness, the evaluation of **software as a medical device (SaMD)**, and post-market surveillance of connected health systems. These efforts are designed to ensure that digital medical products meet both performance and safety benchmarks, particularly as they undergo updates and evolve in clinical environments [120,121,122,123].

Similarly, within the European Union, the **Medical Device Regulation (MDR)** enforces strict requirements on the cybersecurity risk assessment and **data integrity management** of all **connected Class II and Class III devices**. These regulations aim to protect patient safety, ensure device reliability, and prevent unauthorized access to sensitive medical data, all while facilitating innovation in wearable, implantable, and ingestible health technologies [124]. In both regions, manufacturers are expected to comply with internationally recognized clinical evaluation standards—such as **ISO 14155**, [125] which governs the design and execution of **clinical trials for medical devices**—to demonstrate the efficacy and safety of their products in real-world applications.

Regulatory approval also depends on accurate **risk classification** of each device, taking into account its level of invasiveness, intended function, and interaction with biological systems. For example, wearable devices that passively monitor health parameters are categorized differently from closed-loop systems that actively intervene in physiological processes. In addition, developers must ensure **interoperability** with existing healthcare infrastructures. This typically involves adherence to data exchange protocols such as **Health Level Seven (HL7)** and **Fast Healthcare Interoperability Resources (FHIR)**, which are essential for seamless integration into **electronic health records (EHRs)** and other clinical information systems.

Beyond compliance, there is a growing consensus that **ethical design principles** must be embedded into the development lifecycle of IoB devices. This includes preserving **user autonomy**, enabling **fail-safes**, and incorporating **manual overrides**, particularly in systems capable of autonomous or semi-autonomous intervention—such as **closed-loop insulin delivery systems**, **neuromodulators**, or **deep brain stimulation (DBS)** implants [126,127,128]. These features are not only critical for patient safety, but also for fostering trust and acceptance of IoB technologies in both clinical and consumer contexts [129].

## 7. Future Perspectives and Open Challenges

The convergence of MEMS, BioMEMS, and IoB technologies has already reshaped the landscape of personalized healthcare—but this transformation is only beginning. As these systems become more autonomous, intelligent, and embedded within the human body, new frontiers and challenges emerge across science, engineering, and ethics. The IoB represents a transformative frontier in personalized medicine, enabling continuous, context-aware monitoring and adaptive therapeutic intervention. However, several key scientific and technological challenges must be addressed to fully realize this potential.

A foremost challenge is **long-term biocompatibility and stability**. Despite significant advances in biomaterials, many implantable and ingestible devices still face degradation, inflammatory responses, or encapsulation over time. Research into bioresorbable materials, anti-fouling surface coatings, and dynamic interfaces capable of long-term physiological integration will be critical.

**Energy autonomy** is another central bottleneck. Most IoB devices rely on batteries, which limit operational lifetimes and require invasive replacement procedures. Future systems must integrate hybrid energy-harvesting solutions—combining triboelectric, piezoelectric, thermoelectric, and biofuel cell technologies—with ultra-low-power electronics and duty-cycled operation to achieve self-sustaining functionality.

The **security, privacy, and scalability** of IoB networks also present significant research opportunities. As device numbers grow, robust encryption and authentication mechanisms must be developed that do not compromise power efficiency. Additionally, edge–cloud architectures capable of handling high-throughput physiological data while preserving patient privacy will be essential.

Finally, the incorporation of **artificial intelligence (AI)** and **machine learning (ML)** into IoB ecosystems will be transformative. AI-driven edge analytics can enable real-time anomaly detection, predictive diagnostics, and autonomous therapeutic decisions, moving IoB from reactive monitoring to proactive healthcare systems. Future research should focus on explainable AI models tailored to biomedical data and federated learning approaches that safeguard privacy.

Addressing these challenges requires interdisciplinary collaboration across **materials science, microfabrication, wireless communication, embedded AI, and regulatory science**. As these advances converge, the IoB paradigm will evolve from proof-of-concept devices into robust clinical platforms, supporting precision medicine and enabling continuous, patient-centered care across diverse healthcare environments.

### 7.1. Emerging Trends in IoB-Enabled Healthcare

The next generation of IoB systems is being shaped by rapid advances in enabling technologies, including artificial intelligence (AI), autonomous power systems, and adaptive materials. These trends are driving a transition from static sensing platforms to intelligent, self-sustaining, and context-aware microsystems capable of continuous monitoring and real-time decision-making. At the forefront of these developments are AI-assisted signal processing, self-powered devices based on energy-harvesting transducers, and biointegrated materials designed for long-term implantation. Together, these innovations point toward an ecosystem of IoB devices that will become increasingly autonomous, predictive, and clinically actionable. One of the most significant directions is the development of **multi-modal sensing platforms**. Future devices will integrate mechanical, electrical, biochemical, and thermal sensors on a single microfabricated substrate, enabling the **simultaneous capture of diverse physiological signals**. This convergence allows for improved diagnostic accuracy through **sensor fusion**, enhancing contextual awareness in real-time. Such integration is especially valuable in the continuous management of **chronic diseases** such as diabetes, cardiovascular conditions, or neurodegenerative disorders, where multiple biomarkers must be interpreted together to assess patient status effectively [130,131,132].

Another powerful enabler of next-generation IoB systems is **artificial intelligence**. The incorporation of **machine learning algorithms** trained on **longitudinal biometric datasets** enables **predictive analytics** capable of anticipating critical events such as arrhythmias, respiratory failure, or metabolic imbalances. By continuously learning from real-time and historical health data, AI-enhanced IoB platforms can tailor therapeutic responses, flag deviations from individual baselines, and support proactive, rather than reactive, medical intervention. These capabilities will be essential for the transition from episodic care to **continuous, personalized health management** [133,134].

Power supply limitations have long constrained the usability of implantable and long-term wearable devices. To overcome this, research is advancing toward self-powered and energy-harvesting MEMS architectures. These systems integrate thermoelectric generators (TEGs), piezoelectric films, and biofuel cells that scavenge energy from physiological sources such as body heat, motion, and internal biochemical reactions. Eliminating the need for periodic battery replacement significantly improves device autonomy, patient safety, and comfort, particularly for implants that are not easily accessible once deployed [135,136].

In tandem with energy autonomy, there is a growing emphasis on **flexible and biodegradable electronics**. Advances in materials science now allow the fabrication of **stretchable, skin-conformal devices**, including **electronic tattoos** and **transient implants** made from materials such as silk, cellulose, and PLA. These platforms reduce immune responses, conform to curved and dynamic body surfaces, and—when intended for temporary use—**dissolve harmlessly** in the body after serving their diagnostic or therapeutic function, eliminating the need for surgical removal [137,138,139,140,141].

Finally, the increasing volume and resolution of physiological data collected by IoB devices are laying the groundwork for the implementation of **digital twins**—high-fidelity, individualized virtual models that replicate a patient’s physiological, metabolic, and anatomical characteristics. These digital counterparts can be used to simulate treatment responses, optimize drug regimens, and conduct **virtual clinical trials**, greatly accelerating personalized therapy development and enabling precision risk assessment for disease progression or treatment failure. As digital twins become more integrated with IoB data streams and AI inference engines, they are poised to **transform the landscape of virtual healthcare and clinical decision support** [142,143,144,145].

### 7.2. Persistent Scientific and Technical Challenges

Despite the rapid advancement and evident potential of IoB-integrated personalized healthcare systems, several **critical barriers** continue to hinder their widespread adoption and long-term functionality. One of the foremost challenges lies in ensuring **biocompatibility and long-term in vivo stability**. Many MEMS and BioMEMS materials, though initially functional, are prone to **degradation**, **corrosion**, or **biofouling** once implanted. Inflammatory responses may lead to fibrous tissue encapsulation, which attenuates signal transmission, disrupts sensor calibration, or impairs actuator performance over time. As a result, maintaining consistent and accurate measurements across extended operational periods remains a substantial obstacle [146,147,148,149].

Equally complex is the **trade-off between miniaturization and device functionality**. While reducing device size is essential for patient comfort, minimally invasive deployment, and integration into microenvironments such as vascular systems or neural tissue, excessive miniaturization often compromises **power availability**, **data throughput**, and **signal-to-noise ratios**. This imposes design constraints on sensing resolution, wireless communication modules, and onboard computational capacity, especially in the case of autonomous, implantable IoB nodes [150,151]. In parallel, the lack of **interoperability and data standardization** across device manufacturers, healthcare providers, and electronic health systems presents a systemic limitation. The absence of universally accepted **data formats**, **security protocols**, and **communication interfaces** complicates unified integration into broader digital health infrastructures. This fragmentation not only impairs cross-platform compatibility but also increases the risk of cybersecurity vulnerabilities and hinders real-time clinical decision-making [152,153,154].

Beyond technical difficulties, **clinical translation and user acceptance** remain key determinants of IoB success. Even the most sophisticated systems must undergo **regulatory validation**, achieve **cost-effectiveness**, and demonstrate **high user-friendliness** to be trusted by physicians and adopted by patients. Factors such as intuitive interfaces, non-invasiveness, reliability, and alignment with existing clinical workflows play crucial roles in ensuring long-term engagement. Finally, as IoB technologies increasingly intersect with **autonomous decision-making** and **AI-driven interventions**, a new set of **ethical dilemmas and societal readiness questions** emerges. Core issues include **data ownership**, **consent dynamics**, **algorithmic transparency**, and **the right to override or refuse machine-led actions**. For instance, who ultimately holds authority over an implantable system capable of delivering therapeutic stimuli or altering physiological states? Such concerns challenge traditional notions of bodily autonomy and demand the development of ethical frameworks that can keep pace with the capabilities of intelligent, embodied medical technologies.

### 7.3. Research Directions and Collaborative Opportunities

While the technological trends outlined in Section 7.1 define the future direction of IoB systems, their realization depends on overcoming several fundamental challenges. Research efforts are increasingly focused on developing specific solutions that address these barriers. Examples include **novel bioresorbable materials** to improve long-term biocompatibility, **lightweight cryptographic protocols** to ensure secure data transmission without excessive power consumption, and **hybrid energy-harvesting systems** that combine triboelectric and piezoelectric elements for continuous autonomous operation. Progress in these areas will be critical for translating emerging technologies into robust, clinically deployable IoB platforms. These next-generation platforms offer enormous potential for **short-term implants**, **pediatric applications**, and **post-operative monitoring**.

In parallel, the integration of **artificial intelligence** with MEMS platforms and **on-demand drug delivery systems** could give rise to **closed-loop therapeutic architectures**—systems that detect pathological conditions, interpret them autonomously, and administer appropriate treatment without external intervention. Such devices, for instance, could detect seizures or arrhythmias and trigger neural stimulation or pharmaceutical release accordingly, representing a significant leap toward real-time, adaptive healthcare. Ensuring that such intelligent systems remain **trustworthy**, **secure**, and **transparent** will require robust **open-source frameworks** for ethical data governance. This includes not only encryption and privacy-by-design but also **auditability**, **user control**, and **bias mitigation mechanisms** in AI models used for diagnosis or therapy optimization. Publicly accessible, peer-reviewed platforms for algorithm training, validation, and verification can help establish **community standards** and **regulatory confidence**. Furthermore, comprehensive **longitudinal studies** across **diverse demographic groups and clinical environments** are crucial for evaluating the long-term safety, reliability, and equity of IoB devices. Such studies can reveal how factors such as age, comorbidities, lifestyle, or environmental conditions influence device performance, adherence, and outcomes, helping to shape inclusive design strategies and clinical protocols.

Finally, the international nature of connected health ecosystems necessitates **global regulatory harmonization**. Standardized procedures for **cross-border data sharing**, **device certification**, and **telemedicine practices** will be essential for enabling IoB devices to function seamlessly in multinational clinical trials and global healthcare systems. This alignment will help accelerate innovation while safeguarding patient rights and public health across jurisdictions. Bridging the gap between technological promise and clinical deployment requires coordinated research across materials science, microfabrication, data security, and AI-driven analytics. Addressing these challenges through targeted research pathways will pave the way for next-generation IoB systems that are not only more intelligent and autonomous but also safe, reliable, and compliant with clinical standards.

## 8. Conclusions

The convergence of Micro-Electro-Mechanical Systems (MEMS), BioMEMS, and the Internet of Bodies (IoB) has opened an unprecedented frontier in personalized healthcare. From continuous glucose monitors and wearable ECG patches to ingestible sensors and smart prosthetics, these technologies transform the body into a connected data platform capable of real-time diagnostics, adaptive therapy, and remote clinical oversight.

This review has traced the **evolution of MEMS technologies** from early micromechanical structures to highly integrated, multifunctional biomedical platforms. It was outlined the development and impact of **BioMEMS** in minimally invasive diagnostics and treatment, and illustrated how these technologies are now embedded within the **IoB architecture**, enabling a shift toward predictive, personalized medicine. By classifying IoB devices into non-invasive, invasive, and incorporated categories, we highlighted their diverse functions, risks, and regulatory needs.

Key use cases were explored in the context of diabetes, cardiovascular disease, neurology, and mobility enhancement, showing how these technologies are already reshaping clinical care. However, alongside opportunity lies complexity: **cybersecurity threats**, **data privacy concerns**, **interoperability gaps**, and **ethical dilemmas** pose significant challenges to sustainable implementation.

Looking ahead, the future of healthcare lies in intelligent, decentralized systems that respond to each patient’s unique physiological signature. Realizing this vision will require interdisciplinary collaboration across engineering, clinical science, materials research, data science, and policy. As these domains continue to converge, IoB-enabled MEMS and BioMEMS technologies are poised to become foundational to the next generation of healthcare—one that is smarter, safer, and truly personalized.

## Figures and Tables

**Figure 1 micromachines-16-01182-f001:**
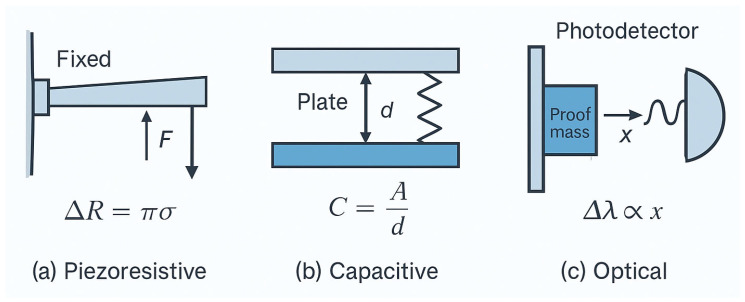
Transduction mechanisms of MEMS: (**a**) piezoresistive beam bending and resistance change; (**b**) capacitive parallel-plate displacement sensor; (**c**) optical cavity displacement detection.

**Figure 2 micromachines-16-01182-f002:**
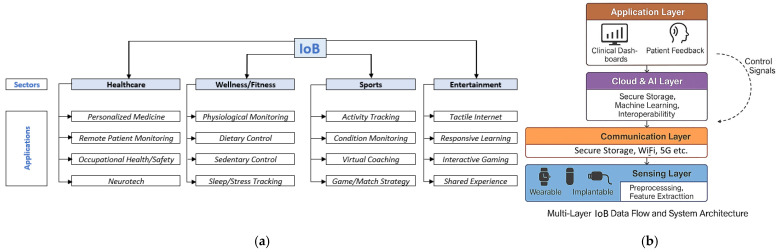
(**a**) Data flow within the Internet of Bodies (IoB) framework. The IoB ecosystem integrates diverse physiological and behavioral data originating from healthcare, wellness, fitness, sports, and entertainment sectors. Adapted from Celik et al. [5]. (**b**) Schematic representation of the multi-layer data flow in the Internet of Bodies (IoB) ecosystem. MEMS and BioMEMS sensors acquire physiological, biochemical, and environmental data, which undergo on-device preprocessing before transmission via wireless communication networks. Cloud-based platforms perform large-scale analytics and integrate the results into clinical decision support systems. The feedback loop enables closed-loop therapeutic control, ensuring adaptive and personalized medical interventions. Adapted and expanded from data presented in [5,6,22].

**Figure 3 micromachines-16-01182-f003:**
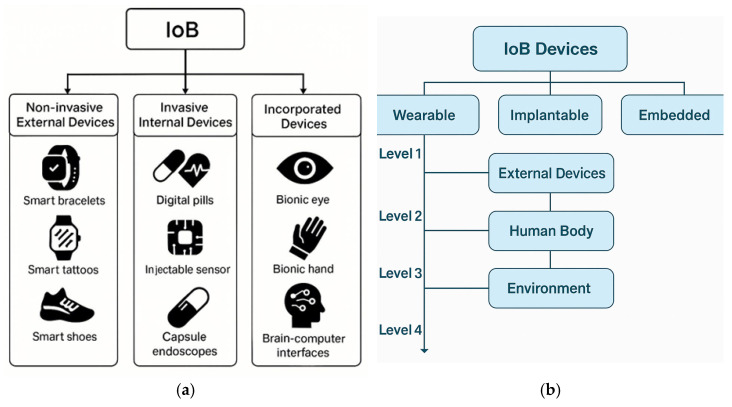
(**a**) Conceptual representation of the Internet of Bodies (IoB) ecosystem, illustrating the classification of connected medical devices based on their interaction with the human body. The diagram highlights three main categories: non-invasive external devices (e.g., wearable sensors and smart textiles), invasive internal devices (e.g., implants, ingestibles), and incorporated devices (e.g., bionic prosthetics, brain–computer interfaces), along with examples and their typical healthcare applications. This layered approach emphasizes the spectrum of bodily integration and the growing convergence between MEMS, BioMEMS, and IoT technologies in personalized medicine (**b**) Classification of IoB devices based on depth of interaction with the human body and directionality of data flow. Non-invasive devices collect physiological or biochemical data externally, invasive devices operate within the body to monitor and intervene in physiological processes, and incorporated systems achieve seamless bidirectional communication with neural or organ-level networks. Examples and typical communication pathways are indicated for each class. Adapted from and expanded based on [Kim et al., Nano-Micro Lett., 2023] and [Shahzad et al., Sensors, 2024] [55,56].

**Figure 4 micromachines-16-01182-f004:**
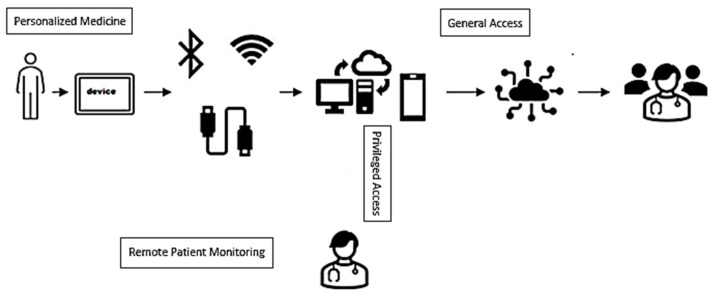
Schematic representation of the IoB data acquisition and processing pipeline. Sensors acquire patient-specific data which is pre-processed locally, transmitted to secure cloud infrastructures, and shared selectively with authorized medical professionals. This loop enables real-time feedback and personalized healthcare decisions.

**Figure 5 micromachines-16-01182-f005:**
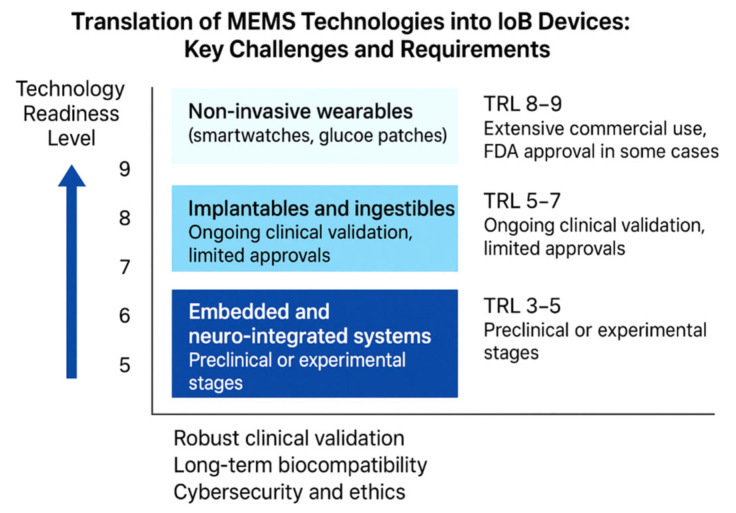
Schematic representation of TRL for MEMS technologies into IoB devices.

**Table 1 micromachines-16-01182-t001:** Comparative benchmarking of representative MEMS and BioMEMS platforms by sensing modality, resolution, size, materials, signal transduction, power requirements, sensitivity, and TRL.

**Platform**	Examples	Sensing Type	Resolution	Size	Materials	Signal Transduction	Power Req.	Sensitivity Range	TRL	Clinical Status	Ref.
MEMS magnetic sensor	Resonant piezoelectric magnetometer	Magnetic field	nT-level	µm-scale beam	Silicon/Magnetostrictive	Piezoelectric resonance	µW–mW	0.1–1 nT	6–8	Preclinical validation	[26]
MEMS optomechanical accelerometer	Photonic-crystal cavity type	Acceleration	~10 µg/√Hz	mm-scale chip	Silicon + photonic crystal	Optical microcavity readout	sub-mW (laser)	5–20 µg/√Hz	5–6	Lab prototype	[27]
Graphene NEMS accelerometer	Suspended graphene ribbon	Acceleration	~µg/√Hz	Nanoscale	Graphene + Si proof mass	Piezoresistive graphene strain sensor	µW electronic bias	0.5–2 µg/√Hz	4–5	Experimental	[28]
BioMEMS implant electrode	Michigan/Utah neural probes	Neural signals	Single-unit spikes (µV-level)	Tens of µm	Si + Pt/Ir metals	Electrophysiological electrodes	µW–mW (wireless IC)	20–200 µV	7–9	Clinical/research use	[29]
BioMEMS microfluidic sweat sensor	Wearable sweat chip	Chemical biomarkers	mM-level ions	mm–cm	PDMS/PET + CNT + electrodes	Electrochemical (amperometric/potentiometric)	µW–mW	1–10 mM	5–7	Pilot trials	[30]

**Table 2 micromachines-16-01182-t002:** Summary of Internet of Things (IoT) applications across diverse domains, illustrating the breadth of data-driven systems. The IoB constitutes a human-centric branch of this expanding digital ecosystem, focused on physiological data collection and healthcare optimization.

Domain	Content	Benefits
**Space** [42,43]	- Non-terrestrial networks (NTN) - Inter-planetary network (IPN) - Aerial network	- The network broadcasts assistance information to facilitate mobility. - Innovative bidirectional storage transfer series was used to correlate various time intervals and aid in maximizing flow in Inter-planetary network
**Sea** [44,45]	- Underwater sensors - Autonomous underwater vehicles (AUV)	- Ocean data analytics - Communicating with each other, collecting data, and transmitting to control centers above the surface at regular internet speeds.
**Underground** [46]	- Agriculture approaches	- Underground data for real-time soil sensing and monitoring
**Industrial** [47,48,49,50]	- Interconnected sensors, instruments, and other devices networked together with computers’ industrial	- Improve and sustain life for everyone on the planet
	- Industry 4.0	- Hybridization of old technologies and new approaches - Ease of production and efficiency - Security standards, and legal scrutiny.
**Defense and public safety** [51]	- Alarms and gas leak detectors	- Public safety
**Medical** [52,53,54,55]	- Gait and waist motions - Wearable antennas - Fall detection sensors - Wheelchairs equipped with IoT sensors	- Cost-effective human–machine interface to enhance the immersion during the long-term rehabilitation. - Online immediate physician response to cure and improve patients’ health - Online human behaviors recognition and judgment under collected data.
**Automotive**	- Reinventing the automobile by enabling connected cars	- Preheating the car - Cars can book their own service appointments

**Table 3 micromachines-16-01182-t003:** Comparative summary of two state-of-the-art smart insole platforms integrating MEMS/nano-sensor arrays and wireless telemetry. The self-powered smart insole [25] demonstrates full device autonomy and durability, whereas the flexible smart insole [56] prioritizes ultra-high spatial resolution via screen-printed sensors.

Platform/Device	Sensor Architecture	Clinical Focus & Validation	Power & Communication	Key Advantages	Main Limitations
**E-Vone Smart Shoes** (commercial)	MEMS pressure sensors integrated into insole; inertial motion unit (IMU)	Fall detection and emergency alerting in elderly users; pilot clinical trials in retirement home settings	Rechargeable Li-ion battery (~5 days); Bluetooth Low Energy (BLE)	Commercial availability, robust wireless connectivity, simple user interface	Limited biomechanical parameter analysis; not suitable for diagnostic gait assessment
**Sensoria Smart Footwear** (commercial)	Textile-integrated pressure sensors + 3-axis accelerometer	Step counting, gait pattern monitoring, rehabilitation support; validated in post-stroke gait studies	Rechargeable battery (~7 days); Bluetooth	Comfortable wearable design, clinically validated gait metrics	Limited spatial resolution; battery replacement needed for continuous clinical use
**Platform/Device**	Sensor Architecture	Clinical Focus & Validation	Power & Communication	Key Advantages	Main Limitations
**Self-powered Smart Insole** [25]	Nonlinear synergistic pressure sensor + IMU	Real-time gait monitoring and plantar pressure mapping (22 sensors) over daily use; validated via smartphone app & compression testing (>180,000 cycles)	Self-powered via flexible solar + wireless transmission (Bluetooth)	Full autonomy, durability, spatial mapping, real-time visualization	Field robustness not yet fully tested; reliance on solar harvesting
**Flexible Smart Insole** [56]	Screen-printed nanomaterial piezoresistive sensor array (173 sensors) + integrated electronics	Daily plantar pressure monitoring; human subject trials under walking tasks	Battery-powered + Bluetooth	High spatial resolution, scalable printing method, flexibility	Power/battery lifetime & external validation still limited

**Table 4 micromachines-16-01182-t004:** Representative MEMS and IoB devices used in personalized healthcare by application area.

Application Domain	Device Type	Example Functionality	Sensing Mechanism	Invasiveness	Connectivity
Diabetes	CGM, Smart Insoles	Glucose monitoring, pressure mapping	Optical, pressure, thermal	Invasive/Non	BLE/Wi-Fi
Cardiology	Pacemaker, ECG Patch	HR monitoring, pacing, arrhythmia alert	Electrical, acoustic	Invasive	RF, X-band
Respiratory Disorders	Smart Bracelets, Masks	SpO_2_, respiratory rate, temperature	Optical, triboelectric	Non-invasive	BLE, NB-IoT
Neurology	E-tattoos, Ear-EEG	EEG, hydration, seizure detection	Electrical, bioimpedance	Non-invasive	Bluetooth, Cloud
Mobility/ Orthopedics	Smart Shoes, Insoles	Gait monitoring, fall detection	IMU, pressure sensors	Non-invasive	App-integrated

**Table 5 micromachines-16-01182-t005:** Classification of IoB healthcare devices by body interaction modality with examples.

Device Type	Examples	Characteristics	Key Challenges
**Non-Invasive**	Smart bracelets/watches, e-tattoos, smart shoes	Wearable, flexible, user-friendly, low risk	Signal stability, skin contact
**Invasive—Internal** **[86,87,88]**	Pacemakers, digital pills, ingestible sensors, capsule endoscopes, chips buried under the skin	Implanted or ingested, real-time internal data	Biocompatibility, infection, surgery
**Incorporated** **[89,90,91,92]**	Bionic eyes, bionic hand, brain–machine interfaces	Deep integration, bidirectional data flow	Ethical risk, neural interfacing, power

## Data Availability

Data are contained within this paper.

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
