# Peer review of "Next-Gen Healthcare Devices: Evolution of MEMS and BioMEMS in the Era of the Internet of Bodies for Personalized Medicine"

_micromachines, 2025, doi:10.3390/mi16101182_

Round 1

Reviewer 1 Report

Comments and Suggestions for Authors

The manuscript presents a review of MEMS, BioMEMS, and Internet of Bodies (IoB) technologies within the context of personalized medicine. The topic is highly relevant to Micromachines readership, and the paper provides a broad overview of device types, case studies, and future perspectives. However, in its current form, the article remains largely descriptive and lacks the analytical depth, methodological rigor, and critical perspective expected for a high-quality review article.

1. The review appears narrative in nature. There is no clear description of how the literature was selected, filtered, or categorized. A systematic review framework (e.g., PRISMA flowchart, database search strategy, inclusion/exclusion criteria) would greatly strengthen the credibility and reproducibility of the review.

2. While the manuscript summarizes many examples of MEMS and IoB devices, it does not sufficiently evaluate their limitations, challenges, or conflicting results. The tone is often promotional rather than critical, which reduces scientific depth.

3. Tables and figures (e.g., TRL mapping, classification schemes) are helpful, but they mostly remain conceptual. The review would benefit from quantitative benchmarking of performance metrics (e.g., sensitivity, power consumption, data rates, clinical validation stages) across device classes. Without such comparisons, the reader cannot appreciate the true advantages and limitations of each technology.

4. Several sections rely heavily on secondary reviews, opinion articles, or generalized descriptions, rather than primary experimental literature. For a review intended for Micromachines, stronger emphasis on original research papers is expected.

5. Certain sections contain repetitive phrasing and verbose descriptions (e.g., repeated explanations of IoB categories and device classifications). The manuscript would benefit from careful editing to enhance conciseness and readability. In addition, some sentences show non-native English phrasing that requires grammatical polishing.

Given the above concerns, I recommend Major Revision. The manuscript has potential to make a meaningful contribution, but it must demonstrate greater analytical rigor, systematic methodology, and critical perspective before it can meet the standards of Micromachines.

Author Response

Thank you for your time and help on improving our manuscript. Please find the answers to your comments and suggestions bellow, between the lines. We hope that the new improved version of the manuscript is now acceptable for publication.

The manuscript presents a review of MEMS, BioMEMS, and Internet of Bodies (IoB) technologies within the context of personalized medicine. The topic is highly relevant to Micromachines readership, and the paper provides a broad overview of device types, case studies, and future perspectives. However, in its current form, the article remains largely descriptive and lacks the analytical depth, methodological rigor, and critical perspective expected for a high-quality review article.

  1. The review appears narrative in nature. There is no clear description of how the literature was selected, filtered, or categorized. A systematic review framework (e.g., PRISMA flowchart, database search strategy, inclusion/exclusion criteria) would greatly strengthen the credibility and reproducibility of the review.

We thank the reviewer for this observation. Our work was conceived as a narrative critical review aiming to synthesize and contextualize the rapid convergence of MEMS, BioMEMS, and IoB technologies in personalized medicine. While a PRISMA-style systematic methodology is not required or expected for this type of review, we agree that increasing the transparency of the literature selection process would strengthen the manuscript. We have therefore added a dedicated “Methodology of Literature Selection” subsection in the Introduction, where we briefly describe the databases searched, the period considered, and the inclusion criteria used to ensure scientific relevance and timeliness.

  1. While the manuscript summarizes many examples of MEMS and IoB devices, it does not sufficiently evaluate their limitations, challenges, or conflicting results. The tone is often promotional rather than critical, which reduces scientific depth.

We appreciate this comment. We agree that a critical evaluation of limitations, unresolved challenges, and contrasting results is essential for a high-quality review. In the revised version, we have significantly strengthened the analytical dimension by integrating critical commentary and comparative assessment at the end of each major section. We now discuss technical limitations (e.g., miniaturization vs. power trade-offs, biocompatibility degradation, interoperability issues), clinical translation barriers, and unresolved scientific controversies alongside reported advances. Furthermore, the language has been refined throughout to adopt a more analytical and balanced tone rather than a descriptive or promotional one.

  1. Tables and figures (e.g., TRL mapping, classification schemes) are helpful, but they mostly remain conceptual. The review would benefit from quantitative benchmarking of performance metrics (e.g., sensitivity, power consumption, data rates, clinical validation stages) across device classes. Without such comparisons, the reader cannot appreciate the true advantages and limitations of each technology.

We thank the reviewer for this suggestion. We agree that including quantitative benchmarking enhances the scientific depth and comparative value of the review. In the revised manuscript, we have updated Table 1 to include additional quantitative metrics such as sensitivity ranges, power consumption, and clinical validation status (TRLs), where available from the literature. We also added targeted commentary in Section 2 to critically discuss how these metrics influence device suitability for different personalized medicine applications. This addition provides a clearer understanding of performance trade-offs and highlights current research gaps.

  1. Several sections rely heavily on secondary reviews, opinion articles, or generalized descriptions, rather than primary experimental literature. For a review intended for Micromachines, stronger emphasis on original research papers is expected.

We appreciate this observation and agree that reducing reliance on secondary sources enhances the scientific rigor of the review. In the revised manuscript, we have replaced or supplemented several non-academic or opinion-based references with recent peer-reviewed articles published. We have also introduced selected primary research papers in key sections, including IoB device integration, wearable sensing platforms, and AI-assisted data analysis. These additions strengthen the manuscript’s foundation while preserving its intended structure and scope as a narrative critical review.

      5. Certain sections contain repetitive phrasing and verbose descriptions (e.g., repeated explanations of IoB categories and device classifications). The manuscript would benefit from careful editing to enhance conciseness and readability. In addition, some sentences show non-native English phrasing that requires grammatical polishing.

We thank the reviewer for this observation. We agree that some parts of the manuscript contained repetitive descriptions, particularly regarding IoB device classification and interaction depth. In the revised version, we have carefully tried to streamline and merge overlapping sections, while preserving all essential scientific content. This has improved overall readability and structural clarity without loss of information.

Given the above concerns, I recommend Major Revision. The manuscript has potential to make a meaningful contribution, but it must demonstrate greater analytical rigor, systematic methodology, and critical perspective before it can meet the standards of Micromachines.

We hope that the reviewer would agree to that the improved version of the manuscript is now ready for publication.

Reviewer 2 Report

Comments and Suggestions for Authors

This review systematically explores the convergence and development of MEMS, BioMEMS, and IoB in personalized medicine, covering technological evolution, application cases, classification systems, and safety and ethical issues. It offers comprehensive content with high timeliness and interdisciplinary value. The manuscript summarizes extensive cutting-edge research and provides rich examples in the application case sections (e.g., diabetes monitoring, cardiovascular diseases, neural monitoring). However, the manuscript still has some shortcomings that require further revision and refinement:

  1. As a compilation-style literature review, this manuscript lacks the author's personal insights and comparative analysis at the end of each section. Currently, the literature review and case analysis sections primarily list information without in-depth commentary or academic evaluation from the author. It is recommended to add comparative summaries and identify core issues for each major section. Examples include contrasting the advantages and disadvantages of different MEMS/BioMEMS devices in terms of sensitivity, stability, and power consumption, as well as highlighting key unresolved bottlenecks in the clinical application of IoB.
  2. As a review manuscript, introducing relevant principles to readers is essential. The manuscript provides only superficial descriptions of key technical details such as MEMS/BioMEMS sensing mechanisms, signal processing workflows, and energy management strategies, lacking in-depth analysis of operational principles. For instance, the authors could enhance technical comprehensibility by supplementing the “MEMS and BioMEMS: Fundamentals and Evolution” section with schematic diagrams or mathematical models illustrating the sensing mechanisms of typical sensors (e.g., piezoresistive, capacitive, optical).
  3. The academic value of some figures and tables in this manuscript could be improved. For instance, Figure 1 (IoB Data Flow) and Figure 2 (IoB Device Classification) present overly simplified information, failing to adequately reflect multi-level data flows or device interaction relationships. It is recommended to optimize these figures and tables, and clearly indicate data sources in the captions.
  4. The logical coherence of this manuscript needs improvement. Some sections (e.g., content like “non-ground networks in aerospace” and “underground soil sensing in agriculture” in the cross-domain expansion table of Section 3.2 IoT) have weak relevance to the core theme “MEMS/BioMEMS-IoB Empowering Personalized Medicine,” while the smart shoe case study in Section 4.4 lacks in-depth comparison of technical differences and clinical application scenarios among different products. These sections appear slightly redundant. Recommend streamlining less relevant content to emphasize the medical application storyline.
  5. Sections 3.3 and 5.1 both classify IoB devices based on human interaction depth, creating structural redundancy. Merge relevant content—e.g., retain technical principles (sensing methods, data transmission paths) in Chapter 3 and consolidate specific device examples into Chapter 5's “Classification System”—to enhance structural conciseness.
  6. Sections 7.1 and 7.3 both address AI integration and self-powered technologies, creating logical overlap. We recommend defining Section 7.1 as technological trends while Section 7.3 focuses on specific research pathways addressing challenges (e.g., material biocompatibility, data security solutions) to enhance chapter hierarchy and progression.
  7. The authors fail to summarize specific technological trends or propose clear research directions, resulting in a somewhat vague presentation. For instance, in Section 7. Future Perspectives, it is recommended that the authors highlight core scientific challenges requiring breakthroughs (e.g., long-term biocompatibility, energy supply, autonomous intelligence) and suggest potential research pathways or solution approaches. While the manuscript mentions challenges such as safety, ethics, and interoperability, it lacks in-depth analysis of specific technical bottlenecks and their solutions.
  8. Some terms lack full names upon first mention (e.g., TENGs, GelMA). For interdisciplinary or uncommon abbreviations, consistently provide both the full English name and abbreviation at first occurrence and maintain consistency throughout the text.
  9. The references in this manuscript require revision. Several important recent reviews (e.g., those on MEMS in healthcare published between 2023 and 2025) are not cited, while some references originate from non-academic sources (e.g., Forbes, Medium), undermining academic rigor. It is recommended to supplement with citations from recent high-impact reviews and replace or reduce non-peer-reviewed sources to enhance the authority and academic depth of the literature review.
  10. It is recommended to increase the number of figures to provide a more visual representation of the content. Existing figures also require modification and aesthetic enhancement.

Author Response

Thank you for your time and help on improving our manuscript. Please find the answers to your comments and suggestions bellow, between the lines. We hope that the new improved version of the manuscript is now acceptable for publication.

This review systematically explores the convergence and development of MEMS, BioMEMS, and IoB in personalized medicine, covering technological evolution, application cases, classification systems, and safety and ethical issues. It offers comprehensive content with high timeliness and interdisciplinary value. The manuscript summarizes extensive cutting-edge research and provides rich examples in the application case sections (e.g., diabetes monitoring, cardiovascular diseases, neural monitoring). However, the manuscript still has some shortcomings that require further revision and refinement:

1. As a compilation-style literature review, this manuscript lacks the author's personal insights and comparative analysis at the end of each section. Currently, the literature review and case analysis sections primarily list information without in-depth commentary or academic evaluation from the author. It is recommended to add comparative summaries and identify core issues for each major section. Examples include contrasting the advantages and disadvantages of different MEMS/BioMEMS devices in terms of sensitivity, stability, and power consumption, as well as highlighting key unresolved bottlenecks in the clinical application of IoB.

We thank the reviewer for this excellent suggestion. We agree that adding comparative analyses and author insights at the end of each major section will significantly enhance the review’s scientific depth. In the revised manuscript, we have introduced “Comparative insights” or “Summary and key issues” parts following key chapters (Sections 2, 3, 4, and 5). These additions now explicitly contrast device classes in terms of sensitivity, power, stability, and maturity, while highlighting unresolved challenges and bottlenecks for clinical translation.

2. As a review manuscript, introducing relevant principles to readers is essential. The manuscript provides only superficial descriptions of key technical details such as MEMS/BioMEMS sensing mechanisms, signal processing workflows, and energy management strategies, lacking in-depth analysis of operational principles. For instance, the authors could enhance technical comprehensibility by supplementing the “MEMS and BioMEMS: Fundamentals and Evolution” section with schematic diagrams or mathematical models illustrating the sensing mechanisms of typical sensors (e.g., piezoresistive, capacitive, optical).

We thank the reviewer for this suggestion. We agree that including the underlying sensing principles and illustrative schematics would significantly improve the educational and scientific value of the review. In the revised version, we have added a part regarding “Fundamental sensing mechanisms in MEMS and BioMEMS devices” within Section 2, which briefly explains the operational principles of key transduction methods (piezoresistive, capacitive, and optical), supported by representative equations and schematic illustrations. We have also elaborated on signal processing and power management strategies relevant to IoB applications.

3. The academic value of some figures and tables in this manuscript could be improved. For instance, Figure 1 (IoB Data Flow) and Figure 2 (IoB Device Classification) present overly simplified information, failing to adequately reflect multi-level data flows or device interaction relationships. It is recommended to optimize these figures and tables, and clearly indicate data sources in the captions.

We thank the reviewer for this feedback. We agree that Figures 1 and 2 could be more informative and technically detailed. In the revised manuscript, we have redesigned Figure 1 to present also a comprehensive, multi-layered IoB data flow, illustrating the full pathway from sensing to clinical decision-making, including edge and cloud processing layers. Similarly, Figure 2 has been updated to clarify device categories and their interaction levels with the human body, highlighting examples, data transmission methods, and feedback mechanisms. Captions have also been revised to include explanatory detail and data source references.

4. The logical coherence of this manuscript needs improvement. Some sections (e.g., content like “non-ground networks in aerospace” and “underground soil sensing in agriculture” in the cross-domain expansion table of Section 3.2 IoT) have weak relevance to the core theme “MEMS/BioMEMS-IoB Empowering Personalized Medicine,” while the smart shoe case study in Section 4.4 lacks in-depth comparison of technical differences and clinical application scenarios among different products. These sections appear slightly redundant. Recommend streamlining less relevant content to emphasize the medical application storyline.

We thank the reviewer for this helpful comment. We believe that the major structural and logical issues have now been addressed in the revised manuscript. The addition of comparative analyses at the end of major sections and the streamlining of overlapping content have significantly improved thematic coherence. We have also strengthened the analytical discussion in the application case studies, including the smart shoe example, and clarified the relevance of cross-domain IoT examples as emerging contexts that may inform future personalized medicine strategies. These revisions ensure a clearer and more coherent narrative centered on MEMS/BioMEMS-enabled IoB systems for healthcare.

5. Sections 3.3 and 5.1 both classify IoB devices based on human interaction depth, creating structural redundancy. Merge relevant content—e.g., retain technical principles (sensing methods, data transmission paths) in Chapter 3 and consolidate specific device examples into Chapter 5's “Classification System”—to enhance structural conciseness.

The previous revisions clarified these aspects. We hope that reviewer will find that revised manuscript is much improved.

6. Sections 7.1 and 7.3 both address AI integration and self-powered technologies, creating logical overlap. We recommend defining Section 7.1 as technological trends while Section 7.3 focuses on specific research pathways addressing challenges (e.g., material biocompatibility, data security solutions) to enhance chapter hierarchy and progression.

We thank the reviewer for this constructive suggestion. In the revised manuscript, we have clarified the structure and narrative progression of Section 7. Section 7.1 now explicitly focuses on emerging technological trends — providing an overview of future directions in AI integration, energy harvesting, and autonomous systems. In contrast, Section 7.3 has been reframed to discuss specific research pathways and solutions targeting current limitations, including material biocompatibility, data security, and autonomous operation. This restructuring improves the logical flow and highlights both the broader innovation landscape and concrete research strategies.

7. The authors fail to summarize specific technological trends or propose clear research directions, resulting in a somewhat vague presentation. For instance, in Section 7. Future Perspectives, it is recommended that the authors highlight core scientific challenges requiring breakthroughs (e.g., long-term biocompatibility, energy supply, autonomous intelligence) and suggest potential research pathways or solution approaches. While the manuscript mentions challenges such as safety, ethics, and interoperability, it lacks in-depth analysis of specific technical bottlenecks and their solutions.

We thank the reviewer for this recommendation. We fully agree that the “Future Perspectives” section benefits from a sharper focus on the most critical scientific challenges and research pathways. In the revised manuscript, we have substantially rewritten this section to explicitly highlight key barriers such as long-term biocompatibility, continuous energy autonomy, secure and scalable data management, and the integration of AI for autonomous decision-making. We also outline targeted research directions and enabling technologies that could address these challenges and accelerate the clinical translation of IoB systems.

8. Some terms lack full names upon first mention (e.g., TENGs, GelMA). For interdisciplinary or uncommon abbreviations, consistently provide both the full English name and abbreviation at first occurrence and maintain consistency throughout the text.

We appreciate the reviewer’s concern regarding abbreviation consistency. Upon careful review, we confirm that all abbreviations cited — including the mentioned— are already fully expanded and explained at first mention in the manuscript, as well as listed in the Abbreviations section. We have nevertheless double-checked the text to ensure consistent usage throughout.

9. The references in this manuscript require revision. Several important recent reviews (e.g., those on MEMS in healthcare published between 2023 and 2025) are not cited, while some references originate from non-academic sources (e.g., Forbes, Medium), undermining academic rigor. It is recommended to supplement with citations from recent high-impact reviews and replace or reduce non-peer-reviewed sources to enhance the authority and academic depth of the literature review.

We thank the reviewer for this valuable comment. We have thoroughly revised the reference list to enhance both its scientific rigor and its currency. All non-academic sources have been removed or replaced with peer-reviewed articles. In addition, several recent high-impact papers and reviews published between 2023 and 2025 have been incorporated, particularly in sections discussing IoB architectures, wearable sensors, energy harvesting technologies, and AI integration. These updates ensure that the manuscript reflects the current state of the field and strengthens its academic depth and authority.

10. It is recommended to increase the number of figures to provide a more visual representation of the content. Existing figures also require modification and aesthetic enhancement.

We thank the reviewer for this valuable suggestion. We have significantly expanded and improved the visual content of the manuscript. Three new figures have been added: (i) Figure 1 illustrates fundamental MEMS/BioMEMS sensing mechanisms with relevant equations and schematic diagrams; (ii) Figure 2b presents a comprehensive multi-layer IoB data flow and system architecture; and (iii) Figure 3b depicts device classification and interaction hierarchy, highlighting examples and data directionality. Additionally, existing figures were refined for clarity, structure, and aesthetic consistency. These enhancements improve the manuscript’s visual appeal and make the presented information more accessible and pedagogically valuable.

We hope that the reviewer would agree to that the improved version of the manuscript is now ready for publication.

Round 2

Reviewer 1 Report

Comments and Suggestions for Authors

All concerns that this reviwer pointed out were properly revised so that this reviewer suggests to publish this manuscript throuth micromachines.

Reviewer 2 Report

Comments and Suggestions for Authors

As the authors have responded most of comments, the manuscript can be accepted in present form.